# A Smart Home Digital Twin to Support the Recognition of Activities of Daily Living

**DOI:** 10.3390/s23177586

**Published:** 2023-09-01

**Authors:** Damien Bouchabou, Juliette Grosset, Sao Mai Nguyen, Christophe Lohr, Xavier Puig

**Affiliations:** 1IMT Atlantique, 44300 Nantes, France; juliette.grosset@ecam-rennes.fr (J.G.); nguyensmai@gmail.com (S.M.N.); christophe.lohr@imt-atlantique.fr (C.L.); 2U2IS, ENSTA Paris, 91120 Palaiseau, France; 3ECAM Rennes, 35170 Bruz, France; 4FAIR, Menlo Park, CA 94025, USA; xavierpuig@meta.com

**Keywords:** smart home, machine learning, home automation, simulator, database, digital twin, transfer learning

## Abstract

One of the challenges in the field of human activity recognition in smart homes based on IoT sensors is the variability in the recorded data. This variability arises from differences in home configurations, sensor network setups, and the number and habits of inhabitants, resulting in a lack of data that accurately represent the application environment. Although simulators have been proposed in the literature to generate data, they fail to bridge the gap between training and field data or produce diverse datasets. In this article, we propose a solution to address this issue by leveraging the concept of digital twins to reduce the disparity between training and real-world data and generate more varied datasets. We introduce the Virtual Smart Home, a simulator specifically designed for modeling daily life activities in smart homes, which is adapted from the Virtual Home simulator. To assess its realism, we compare a set of activity data recorded in a real-life smart apartment with its replication in the VirtualSmartHome simulator. Additionally, we demonstrate that an activity recognition algorithm trained on the data generated by the VirtualSmartHome simulator can be successfully validated using real-life field data.

## 1. Introduction

Over the past few decades, there has been a significant increase in the adoption of smart homes and real-world testbeds, driven by the proliferation of Internet of Things (IoT) devices. These devices enable the detection of various aspects within homes, such as door openings, room luminosity, temperature, humidity, and more. Human Activity Recognition (HAR) algorithms in smart homes have become crucial for classifying streams of data from IoT sensor networks into Activities of Daily Living (ADLs). These algorithms enable smart homes to provide adaptive services, including minimizing power consumption, improving healthcare, and enhancing overall well-being.

Despite the notable advancements in machine learning techniques and the improved performance of HAR algorithms, their practical application to real-world test cases continues to encounter challenges. These challenges primarily stem from the variability and sparsity of sensor data, leading to a significant mismatch between the training and test sets.

### 1.1. A Variable and Sparse Unevenly Sampled Time Series

While HAR based on video data has made significant strides in performance [1], HAR in smart homes continues to encounter specific challenges, as highlighted in the survey by Bouchabou et al. [2]. Recent advances in HAR algorithms, such as convolutional neural networks [3] and fully connected networks [4], along with sequence learning methods like long short-term memory [5], have contributed to advancements in the field. However, the task of recognizing ADLs in a smart home environment remains inherently challenging, primarily due to several contributing factors:Partial observability and sparsity of the data: The input data in HAR consists of traces captured by a variety of sensors, including motion sensors, door sensors, temperature sensors, and more, integrated into the environment or objects within the house [6]. However, each sensor has a limited field of view, resulting in most of the residents’ movements going unobserved by the sensor network in a typical smart home setup. Unlike HAR in videos, where the context of human actions, such as objects of interest or the position of obstacles, can be captured in the images, the sparsity of ambient sensors in HAR does not provide information beyond their field of view. Each sensor activation alone provides limited information about the current activity. For example, the activation of a motion sensor in the kitchen could indicate activities such as “cooking”, “washing dishes”, or “housekeeping”. Therefore, the information from multiple sensors needs to be combined to infer the current activity accurately. Additionally, each sensor activation provides only a partial piece of information about the activity and the state of the environment, unlike videos where both the agent performing the activity and the environment state are visible. Consequently, the time series of sensor activity traces cannot be approximated as a Markov chain. Instead, estimating the context or current state of the environment relies on past information and the relationship with other sensors.Variability of the data: Activity traces between different households exhibit significant variations. The variability arises from differences in house structures, layouts, and equipment. House layouts can vary in terms of apartments, houses with gardens, houses with multiple floors, the presence of bathrooms and bedrooms, open-plan or separate kitchens, and more. The number and types of sensors can also differ significantly between homes. For instance, datasets like MIT [7] use 77–84 sensors for each apartment, while the Kasteren dataset [8] uses 14–21 sensors. The ARAS dataset [9] includes apartments with 20 sensors, while the Orange4Home dataset [10] is based on an apartment equipped with 236 sensors. All these factors, including home topography, sensor count, and their placement, can result in radical differences in activity traces. The second cause of variability stems from household composition and residents’ living habits. ADLs vary depending on the residents’ habits, hobbies, and daily routines, leading to different class balances among ADLs. For example, the typical day of a student, a healthy adult, or an elderly person with frailty will exhibit distinct patterns. Furthermore, the more residents there are, the more the sensor activation traces corresponding to each resident’s activities become intertwined, leading to complex scenarios involving composite actions, concurrent activities, and interleaved activities.

Therefore, it becomes imperative for algorithms to proficiently analyze sparse and irregular time series data to establish effective generalization across a spectrum of house configurations, equipment variations, households with distinct dynamics, and diverse daily habits. It is crucial to recognize that training machine learning algorithms, as well as any other HAR methodologies intended for deployment in these multifaceted contexts, mandates the use of training data that comprehensively encapsulates this extensive variability.

### 1.2. Digital Twins for Generating Similar Data

To bridge the gap between training data and real-world usage data, data generation can be a potential solution, particularly through the concept of digital twins. A digital twin refers to a virtual representation that serves as a real-time digital counterpart of a physical object or process [11,12]. In the context of HAR, a digital twin could be a virtual replica of a target house, complete with the same installed sensors. Within this digital environment, one or multiple avatars can simulate ADLs by modeling the behaviors of residents. In this way, the digital twin can be used to fine-tune algorithms before their deployment in the actual target house.

Moreover, digital twins have the potential to generate data representing a vast range of house configurations, household habits, and resident behaviors, thereby accelerating simulations, facilitating automatic labeling, and eliminating the cost of physical sensors. This extensive dataset can then be utilized for pre-training machine learning models. Furthermore, a digital twin can aid in evaluating the correct positioning and selection of sensors to recognize a predefined list of activities.

Digital twin models have gained significant interest in various application domains, such as manufacturing, aerospace, healthcare, and medicine [13]. While digital twins for smart homes are relatively less explored, digital twins for buildings have been studied extensively. Ngah Nasaruddin et al. [14] define a digital twin of a building as the interaction between the interior environment of a real building and a realistic virtual representation model of the building environment. This digital twin enables real-time monitoring and data acquisition. For example, digital twins of buildings have been utilized in [15] to determine the strategic locations of sensors for efficient data collection.

### 1.3. Contributions

The gap between training and testing data in HAR for smart homes presents significant challenges due to the variability and sparsity of activity traces. In this study, we address this issue by exploring the possibility of generating data suitable for deployment scenarios by using the concept of a smart home digital twin. Our study centers on the application of this method within the domain of HAR deep learning techniques. It is worth highlighting that our proposed method transcends this domain, as its applicability extends seamlessly to encompass both machine learning and non-machine learning approaches.

Our contributions are as follows:We propose a novel approach that paves the way for digital twins in the context of smart homes.We enhance the Virtual Home [16] video-based data simulator to support sensor-based data simulation for smart homes, which we refer to as VirtualSmartHome.We demonstrate, through an illustrative example, that we can replicate a real apartment to generate data for training an ADL classification algorithm.Our study validates the effectiveness of our approach in generating data that closely resembles real-life scenarios and enables the training of an ADL recognition algorithm.We outline a tool and methodology for creating digital twins for smart homes, encompassing a simulator for ADLs in smart homes and a replicable approach for modeling real-life apartments and scenarios.The proposed tool and methodology can be utilized to develop more effective ADL classification algorithms and enhance the overall performance of smart home systems.

In the next section (Section 2), we provide a comprehensive review of the state-of-the-art approaches in HAR algorithms, ADL datasets, and home simulators. Subsequently, in Section 3, we introduce the VirtualSmartHome simulator that we have developed, along with our methodology for replicating real apartments and human activities. Moving forward, in Section 4, we present an evaluation of our simulator, comparing the synthetic data produced by the VirtualSmartHome simulator with real data from a smart apartment. We also demonstrate the potential of our approach by employing the generated datasets for a HAR algorithm.

## 2. Related Work

While recent HAR algorithms have demonstrated improved recognition rates when trained and tested on the same households, their generalizability across different households remains limited. The existing ADL datasets also have their own limitations, prompting the exploration of smart home simulators to generate relevant test data. In this section, we discuss the limitations of current HAR algorithms and ADL datasets, and review the available home simulators.

### 2.1. Machine Learning Algorithms for Activity Recognition Based on Smart Home IoT Data

Numerous methods and algorithms have been studied for HAR in the smart home domain. Early approaches utilized machine learning techniques such as Support Vector Machines (SVM), naive Bayes networks, or Hidden Markov Models (HMM), as reviewed in [17]. However, these models lack generalization and adaptability, as they are designed for specific contexts and rely on hand-crafted features, which are time-consuming to produce and limit the models’ generalization and adaptability.

More recently, deep learning techniques have emerged as a promising approach due to their ability to serve as end-to-end models, simultaneously extracting features and classifying activities. These models are predominantly based on Convolutional Neural Networks (CNN) or Long Short-Term Memory (LSTM).

CNN structures excel at feature extraction and pattern recognition. They have two key advantages for HAR. Firstly, they can capture local dependencies, meaning they consider the significance of nearby observations that are correlated with the current event. Secondly, they are scale-invariant, capable of handling differences in step frequency or event occurrence. For example, Gochoo et al. [18] transformed activity sequences into binary images to leverage 2D CNN-based structures. Singh et al. [19] applied a 1D CNN structure to raw data sequences, demonstrating their high feature extraction capability. Their experiments demonstrated that the CNN 1D architecture yields comparable results to LSTM-based models while being more computationally efficient. However, LSTM-based models still outperform the CNN 1D architecture.

LSTM models are specifically designed to handle time sequences and effectively capture both long- and short-term dependencies. In the context of HAR in smart homes, Liciotti et al. [5] extensively investigated various LSTM structures and demonstrated that LSTM surpasses traditional HAR approaches in terms of classification scores without the need for handcrafted features. This superiority can be attributed to LSTM’s ability to generate features that encode temporal patterns, as highlighted in [20] when compared to conventional machine learning techniques. As a result, LSTM-based structures have emerged as the leading models for tackling the challenges of HAR in the smart home domain.

### 2.2. Activities of Daily Living Datasets

Unfortunately, in order for a deep learning model to achieve sufficient generalization, a large amount of high-quality field data is required.

Several public real home datasets [7,8,9,10,21] are available, and De-La-Hoz-Franco et al. [22] provides an overview of sensor-based datasets used in HAR for smart homes. These datasets primarily utilize binary sensors, such as opening, motion, and pressure sensors, as well as state sensors like temperature sensors. These sensor types are commonly employed in smart home environments due to their battery-powered nature, which simplifies installation and conserves energy. However, this hardware choice results in the generation of event-based time series from these sensors, posing challenges for algorithms. Furthermore, these datasets have certain limitations, including a restricted number of activities and occupants considered, limited sensor usage, and specific types of residents.

The currently available public datasets are insufficient to cover the probability distribution of all possible HAR data and enable algorithms to generalize to application test data.

Moreover, collecting new datasets is a costly and challenging task. For instance, Cook et al. [21] developed a lightweight and easy-to-install smart home design called “a smart home in a box”, aimed at efficiently collecting data once installed in the environment. However, the cost of components for the “smart home in a box” project [21], as shown in Table 1, amounted to USD 2765 in 2013. Although the cost of sensors has decreased over time, there are still challenges in accessing various types of houses and inhabitants. Additionally, collecting data from real inhabitants is time-consuming, and the recording cannot be accelerated like in a simulation. Furthermore, the data requires ground truth, including data stream segmentation (start and end time) and class labels, which necessitates significant user investment and is prone to errors due to manual annotations, as described in [2].

Considering these challenges, researchers suggest generating data using simulation environments.

The objective of simulators in the context of smart homes is to generate simulated data from domestic sensors that accurately reflect real-world scenarios. The production of synthetic data through simulation offers several advantages, including the ability to: (1) collect perfectly controlled ground truth data, and (2) introduce diversity in terms of environments and living habits.

### 2.3. Existing Home Simulators

In the field of HAR, the collection of data and the creation of physical environments with sensors pose significant challenges, including sensor positioning, installation, and configuration. Additionally, the collection of datasets is often limited by ethical protocols and user participation.

Simulation platforms have been widely used in various domains, and there has been a recent surge of interest in developing simulators to represent indoor environments, as highlighted by Golestan et al. [23], see Table 2. Simulation tools offer scalability, flexibility, and extensibility, making them suitable for a wide range of contexts and large-scale applications. They enable rapid, repeatable, and cost-effective prototyping of applications. For example, Bruneau et al. [24] demonstrated that users with basic software development skills can simulate a simple indoor environment in just one hour on average, whereas a similar task would require much more time in the real world.

Some research focuses on providing photorealistic representations of environments to train computer vision models [28,39,40]. Other simulators incorporate actionable objects, allowing agents to learn manipulation and interaction with objects. These advancements have generated growing interest in studying embedded intelligence in home environments and building robots capable of performing household tasks, following instructions, or collaborating with humans at home [41,42,43].

Designing and creating indoor simulation environments is not a straightforward task. Numerous parameters and variabilities that impact daily life must be considered, such as the type and structure of the house and external events like changes in temperature or lighting throughout the day or across seasons. The development of such simulation environments is expensive and time-consuming. As a result, recent research has explored the use of video game platforms that offer sophisticated and successful simulation environments. For instance, Roitberg et al. [27], Cao et al. [44] utilized these environments to generate video data of human activities. However, the use of such environments is limited due to their closed and proprietary nature, making it challenging to incorporate additional functionalities.

Other works have introduced specialized simulation platforms focused on human modeling, providing better control over activities [45]. According to Synnott et al. [46], there are two approaches to designing smart home environment simulators: interaction-based and model-based approaches.

Interactive approaches involve a virtual environment (2D or 3D scenario) where users can act as avatars, interacting with the environment. Most papers in the field adopt this approach for agent modeling because it offers greater variation in agent traces compared to model-driven approaches, especially when a sufficient number of users interact with the simulators.

For example, Park et al. [34] proposed CASS, a simulator that helps designers detect inconsistencies in a defined set of rules related to sensor readings, occupants’ locations, and actuators. Users interact with the simulator through an interface to manipulate the simulated environment.

Buchmayr et al. [36] presented a simulator that models binary sensors (e.g., contact switches, motion, and pressure sensors, temperature sensors) and incorporates faulty sensor behaviors by introducing noise signals to the readings. Users can generate agent traces by interacting with any sensor through a user interface.

Armac and Retkowitz [37] developed a simulator for residential buildings to generate synthetic ADL datasets. The simulator requires designers to define accessible and inaccessible areas (obstacles) and place devices in an indoor environment. Users can interact with virtual agents to produce agent traces by interacting with environmental objects.

Interactive approaches offer accurate and realistic simulations since each action or movement is performed by a real human. However, generating large amounts of data through interactive approaches can be costly and requires significant effort from users. Thus, these approaches are typically suitable for testing single activities or short runs.

On the other hand, model-based approaches involve specifying a reference model for the simulation. The level of abstraction in defining activities determines the precision of the modeled behavior. Kamara et al. [47] argued that this method is sufficient for generating both ADL traces in residential settings and office routines in public buildings.

Bouchard et al. [33] proposed SIMACT, a simulator that allows third-party components to connect to the simulator’s database to receive and store real-time sensor readings. The simulator includes a set of pre-recorded scenarios (agent traces) to ensure data consistency. Users can also define their own scenarios using an XML file. However, SIMACT does not provide a multi-agent environment.

Ho et al. [30] introduced Smart Environment Simulation (SESim), a simulator designed to generate synthetic sensor datasets. The simulator underwent three validation phases: the creation of a smart environment, analysis of the generated data, and training of an activity recognition algorithm using a multi-layer neural network. The algorithm achieved an average recognition accuracy of 83.08% and F1-score of 66.17%. However, the evaluation was conducted solely on synthetic data generated by the simulator.

Lee et al. [31] introduced Persim-3D, a context-driven simulator implemented in Unity 3D. The simulator represents agent activities as sequences of actions and incorporates contexts that determine the conditions under which specific activities can be performed. To assess the external validity of synthetic datasets, the authors compared the data generated by the simulator with real-world data collected from the Gator Tech Smart House (GTSH) [48] and reported an 81% similarity. However, the authors did not evaluate the performance of HAR algorithms using this simulator. Additionally, the current version of the simulator only supports simulation of a single user’s activity.

More recently, hybrid approaches have emerged, combining both model-based and interactive approaches in a single simulator [16,29,32,35]. These approaches offer the advantages of both methods.

Alshammari et al. [29] proposed OpenSHS, a simulator for ADL dataset generation. Designers can use Blender 3D to create the space and deploy devices, and users can control an agent with a first-person view to generate agent traces. The simulator records sensor readings and states based on user interactions. It also supports script-based actions in the environment. However, the project does not appear to be actively updated or used.

Francillette et al. [35] developed a simulation tool capable of modeling the behavior of individuals with Mild Cognitive Impairment (MCI) or Alzheimer’s Disease (AD). The simulator allows the manual control or modeling of an agent based on a behavior tree model with error probabilities for each action. The authors demonstrated that their simulator accurately emulates individuals with MCI or AD when actions have different error probabilities.

Synnott et al. [32] introduced IE Sim, a simulator capable of generating datasets associated with normal and hazardous scenarios. Users can interact with the simulator through a virtual agent to perform activities. The simulator provides an object toolbox with a wide range of indoor objects and sensors, allowing users to create new objects as well. IE Sim collects sensor readings throughout the simulation. The authors demonstrated that the simulator’s data can be used to detect hazardous activities and overlapping activities. IE Sim combines interactive and agent modeling approaches.

Puig et al. [16,49] proposed the Virtual Home simulator, a multi-agent platform for simulating activities in a home. Humanoid avatars represent the agents, which can interact with the environment using high-level instructions. Users can also control agents in a first-person view to interact with the environment. This simulator supports video playback of human activities and enables agent training for complex tasks. It includes a knowledge base that provides instructions for a wide range of activities.

The Virtual Home simulator aligns with our requirements for recognizing activities in a house. Although some common human actions are not yet implemented, such as hoovering or eating, the extensible programming of the simulator allows for modifications. Furthermore, the simulator facilitates the reproduction of human activity scenarios, retrieval of sensor states, and replication of a real smart apartment for a digital twin. It is an ongoing project with an active community.

## 3. Virtual Smart Home: The Simulator

We present Virtual Smart Home, a simulator designed for modeling activities of daily living in smart homes by adapting the Virtual Home simulator [16] to log sensor activations in a smart home environment. To assess its realism, we compare the simulated activities with a multi-user dataset recorded in a real-life living lab.

### 3.1. Design of Virtual Smart Home

After reviewing the available simulators discussed in Section 2.3, we have selected Virtual Home [16] as the foundation for our smart home simulator. Originally developed for computer vision algorithms, Virtual Home is a multi-agent platform designed to simulate activities in a house or apartment. It utilizes humanoid avatars that can interact with their environment and perform activities based on high-level instructions. The simulator incorporates a knowledge base that enables the creation of videos depicting human activities, as well as training agents to perform complex tasks. Additionally, it provides furnished flats for simulation purposes (see Figure 1).

Virtual Home is developed on the Unity3D game engine, which offers robust kinematic, physics, and navigation models. Moreover, users can take advantage of the vast collection of 3D models accessible through Unity’s Assets store, providing access to a diverse range of humanoid models.

Moreover, Virtual Home offers a straightforward process for adding new flats by utilizing the provided Unity project [49]. Each environment in Virtual Home represents an interior flat with multiple rooms and interactive objects. The configuration of each flat scene is stored in a *.json* file, which contains nodes representing each object and their relationships with other objects (specified as *“edge labels”*). For instance, the label *“between”* can be used to describe the relationship between rooms connected by a door. By modifying these description files, users can easily add, modify, or remove objects, enabling the creation of diverse scenes for generating videos or training agents.

Another notable feature of Virtual Home is its capability to create custom virtual databases within specific environments, with a supportive community that contributes new features. In a study conducted by Liao et al. [50], Virtual Home was utilized to generate a novel dataset. The researchers expanded the original Virtual Home database by incorporating additional actions for the agents and introducing programs. These programs consist of predefined sequences of instructions that can be assigned to agents, enabling them to perform activities within their simulated environment.

In Virtual Home, an avatar’s activity is represented by a sequence of actions, such as *“<char0> [PutBack] <glass> (1) <table>”*, as described in [51]. This flexible framework facilitates the training of agents to engage in various everyday activities.

The authors successfully collected a new dataset [50] based on Virtual Home [38], encompassing 3000 daily activities. Furthermore, they expanded the database by incorporating over 30,000 programs, offering a wide range of actions and possibilities. Additionally, the researchers graphed each environment, which consisted of an average of 300 objects and 4000 spatial relationships.

Using Virtual Home, users can create scenarios where 3D avatars perform daily activities, with the ability to capture the simulated actions through a virtual camera. Moreover, the simulator enables the replication of flats, facilitating the creation of digital twins of apartments. However, it is important to note that Virtual Home does not support the acquisition of data through home automation sensors.

In order to enhance the ability to reproduce scenarios and collect data from ambient sensors, we have implemented several new features in our simulator:1.Interactive Objects: While Virtual Home already offers a variety of objects for inclusion in apartments, many are passive and non-interactive. To address this limitation, we added the functionality to interact with some new objects. Agents can now open trash cans, the drawers of column cabinets, and push on toilet faucets. Objects with doors are implemented by splitting them into two parts—one static and one capable of rotation around an axis to simulate interaction. Fluid objects like toilet faucets are simulated by placing the origin point of the fluid at its supposed source.2.Simulation Time Acceleration: To generate a large volume of data quickly, we implemented the ability to accelerate simulation times. This feature utilizes the Time.timeScale function of the Unity game engine. However, the acceleration cannot surpass the rendering time of the Unity game engine, resulting in a maximum four-fold increase in simulation speed.3.Real-Life Apartment Replication and Room Creation: To replicate a real-life apartment, we propose a methodology that involves creating a 2D map of the flat using tools like Sweet Home 3D [52]. This map is then reproduced in Virtual Home, respecting the hierarchical object structure imposed by the simulator. Finally, the interactive objects are placed in a manner similar to their real-world counterparts. We demonstrated the effectiveness of this method by replicating a real-life intelligent apartment based on our room dimension measurements. Additionally, we have introduced the ability to create new rooms, such as outdoor and entrance areas.4.IoT Sensors: While Virtual Home previously focused on recording activities using videos, we have implemented IoT sensors to enhance the simulation. The following sensors have been incorporated: (1) opening/closing sensors, (2) pressure sensors, (3) lights, (4) power consumption, and (5) zone occupancy sensors. Except for the zone occupancy sensors, all other sensors are simulated using the environment graph of the scene. This graph lists all objects in the scene with their corresponding states (e.g., closed/open, on/off). The zone occupancy sensor takes the form of a sensitive floor, implemented using a raycast. It originates from the center of the avatar and is directed downwards. The floor of the flat is divided into rooms, and the intersection with the floor identifies the room in which the avatar is located.5.Simulation Interface: We have developed an interface that allows users to launch simulations by specifying the apartment, ambient sensors, scenarios, date, and time. The interface facilitates the scripting of each labeled activity for reproduction in the simulation. It provides three main functions: (1) the creation of an experiment configuration file, where the simulation flat and desired sensor data can be chosen; (2) the creation of a scenario configuration file, offering choices such as experiment date, simulation acceleration, and various activities with their durations; (3) the association of an experiment configuration file with a scenario configuration file and the subsequent launch of the simulation. This functionality enables the storage of synthetic sensor logs in a database file and provides a comprehensive record of the conducted experiment, including the experiment configuration file and the scenario configuration file.

### 3.2. Assessing the Simulator through Dataset Creation

Our objective is to demonstrate the VirtualSmartHome simulator’s ability to generate realistic data that can be utilized for training HAR algorithms. To achieve this, we undertook the following steps:

Firstly, we recorded ADLs performed by volunteers in our physical smart apartment. This allowed us to collect real-world data that served as the ground truth for our assessment.

Next, we replicated our smart apartment within the VirtualSmartHome simulator, creating a digital twin of the environment. This virtual representation ensured an accurate simulation of the apartment’s layout and characteristics.

Subsequently, we programmed the avatar within the simulator to replicate the ADLs performed by the volunteers. This process involved instructing the virtual agent to mimic the actions and interactions observed during the recording phase. As a result, we generated synthetic sensor logs that mirrored the behaviors of the real-world ADLs.

Finally, we conducted an evaluation to compare the replicated data generated by the simulator with the original real-world data. This assessment aimed to measure the similarity and fidelity of the simulated data, providing insights into the effectiveness of the VirtualSmartHome simulator.

#### 3.2.1. The Smart Apartment

To capture ground truth sensor data, our dataset was created in our smart apartment called Experiment’Haal [53]. This dedicated studio serves as a testbed for experimental assistive devices and enables user testing in a simulated domestic environment. It allows us to validate technological solutions proposed by end-users within their daily life context before deployment on a larger scale.

Our smart apartment is equipped with various smart sensors and a ceiling-mounted fisheye camera. The camera records and observes the ADLs performed by the volunteers, providing visual data for analysis. The smart sensor array includes:1.A sensitive floor that tracks the movement or presence of a person in different areas.2.Motion sensors placed strategically to detect motion in specific locations.3.Magnetic open/close sensors to monitor the status of doors or windows.4.Smart lights that can be controlled remotely or manually adjusted by switches.5.Smart plugs to monitor the status and power consumption of various devices such as the stove, oven, kettle, outlets, and television.6.Pressure sensors installed on the bed and sofa to detect occupancy.

Figure 2 provides a detailed overview of the smart apartment layout and the positions of the sensors.

It is important to note that the sensors in the apartment come from different brands, resulting in heterogeneity and potential communication challenges. To address this, we implemented the xAAL solution [54]. This solution serves as a universal bus and acts as an IP-based smart home hub protocol, enabling us to collect data and events from devices in a unified manner.

To facilitate data collection and analysis, we divided the open space of the apartment into six distinct rooms, as shown in Figure 3. The SensFloor [55], functioning as a presence detector, was utilized to monitor each room. This arrangement allowed us to capture not only the presence of a person in a specific room but also the transitions between rooms.

#### 3.2.2. The Ground Truth Dataset

To generate the ground truth dataset, we recruited a group of volunteers in good health, aged between 18 and 38 years old, consisting of 3 women and 5 men. These volunteers were instructed to participate in three distinct scenarios:1.Morning scenario: This scenario involved getting out of bed and going through a typical morning routine before heading to work.2.Lunchtime scenario: In this scenario, participants returned home from work, prepared and ate lunch, and then went back to work.3.Evening scenario: This scenario focused on participants returning home for the evening until it was time to go to bed.

For these scenarios, we defined a list of 17 possible activity labels, including “Bathe”, “Cook breakfast”, “Cook dinner”, “Cook lunch”, “Dress”, “Eat breakfast”, “Eat dinner”, “Eat lunch”, “Enter home”, “Go to toilet”, “Leave home”, “Read”, “Sleep”, “Sleep in Bed”, “Wash dishes”, and “Watch TV”. These labels were inspired by existing literature datasets such as CASAS [21] and the ADL definitions provided by Katz et al. [56].

We recorded a total of 514 activity sequences with 8 different volunteers across the three scenarios: 159 in the morning, 155 at noon, and 200 in the evening. Table 3, Table 4 and Table 5 show the number of activities performed by the volunteers in each scenario. Table 6 provides a global summary of the generated dataset.

Several post-processing steps were performed, including renaming the sensors and removing certain sensors (e.g., motion sensors, CO_2_ sensors, WiFi, radio level sensors, noise sensor) from the real sequences that could not be implemented in Virtual Home or were not relevant for our project. While our real-life sensors provided power consumption values, we transformed them into *ON* or *OFF* states for simplicity in the virtual dataset, e.g., values of devices such as the TV or the oven.

#### 3.2.3. The Synthetic Dataset

To generate the synthetic dataset, we first recreated our living lab within the VirtualSmartHome simulator. The objective was to create a digital twin of the physical space, ensuring that each object and room was accurately represented with corresponding dimensions. A visual comparison between the real living lab and its virtual representation is shown in Figure 4.

Next, we utilized our implementation of IoT sensors in the VirtualSmartHome simulator to incorporate the sensors present in the living lab. We equipped the different objects in the virtual environment with these sensors. Specifically, we outfitted the floors representing the various rooms with person detection sensors to simulate our sensitive floor (SensFloor) as depicted in Figure 4.

The synthetic dataset was generated by replicating the recorded scenarios from the living lab within the simulator. We scripted an avatar to mimic each action performed by our volunteers. For example, if a volunteer followed these steps for the activity “cooking”: entering the kitchen, washing hands, opening the fridge, closing the fridge, turning on the oven, etc., we scripted the avatars in the VirtualSmartHome simulator to simulate each of these actions. We created one script for each occurrence of an activity performed by our volunteers in all three scenarios, allowing us to obtain a database of scripts that we can reuse later for other environment configurations.

In conclusion, we successfully replicated segments of scenarios involving three randomly selected subjects (subject 3, subject 7, and subject 9). It is noteworthy to mention that the process of scripting actions for the avatars proved to be time-intensive. This aspect also presents a potential avenue for future work, wherein the development of tools to facilitate avatar scripting could enhance the efficiency of the simulation process. Ultimately, we managed to recreate 23 out of 55 scenarios for subject 3, 37 out of 49 scenarios for subject 7, and accomplished full replication for subject 9.

For a comprehensive overview of the synthetic dataset, please refer to the summarized details presented in Table 7.

## 4. Comparison between Synthetic Data and Real Data

To assess the similarity between the synthetic data generated by the VirtualSmartHome simulator and the log sequences obtained from real-life recordings, we first create a sub real-life recordings dataset with the replicated scenarios and we employed two comparison methods.

Firstly, we compared the frequency of sensor activations in the synthetic data with that of the real data. This initial analysis provided insights into the number of sensors triggered during activities and allowed for a comparison between the two datasets.

Secondly, we utilized cross-correlation [57] to measure the similarity between the log sequences from the real-life and synthetic data. Cross-correlation is a statistical measure that calculates the correlation between two sequences at different time lags.

To validate the synthetic log sequences, we implemented an activity recognition algorithm proposed by [5] detailed in the Section 4.3.1. We trained the algorithm using four different tests:1.Training and validating on synthetic data using three subjects with leave-one-out subject validation.2.Training and validating on real data using three subjects with leave-one-out subject validation.3.Training the algorithm on synthetic data from one subject and validating it on the real data of the same subject.4.Training the algorithm on synthetic data from all subjects and validating it on the real data for each subject.

We will present our results in three sections to validate our digital twin approach and draw conclusions regarding activity recognition using synthetic data.

Firstly, in Section 4.1, we will compare the triggered sensor events between the synthetic and real logs. Then, in Section 4.2, we will present the results obtained using cross-correlation to determine the similarity between the sequences of real and synthetic logs. Finally, in Section 4.3, we will showcase the results obtained with the recognition algorithm on the data collected from simulation and reality.

### 4.1. Comparison of Triggered Sensors in Real and Synthetic Logs for Similar Scenarios

To gain an initial understanding of the synthetic data generated, we compared the frequency of sensor activations in the synthetic data with that of the real data.

Figure 5 illustrates the comparison of the number of triggered sensors in the real dataset (in blue) and the synthetic dataset (in red) across 15 scenarios. Most scenarios showed a similar count of triggered sensors in both datasets. However, some scenarios (1, 4, 5, 9, and 13) exhibited an excess of triggered sensors in the synthetic data. Upon examining Table 8, we observed that these scenarios predominantly involved presence sensors, which detect the presence of the avatar in a specific room. The difference in sensor activations can be attributed to the fact that the real-life sensor did not always detect the volunteer, and the path chosen by the avatar in the simulator did not always match the movement of the volunteer during the recording experiment.

In conclusion, the comparison of triggered sensors between the real and synthetic logs for similar scenarios showed a generally close alignment. However, discrepancies were observed in scenarios, in particular for the presence sensors, which can be attributed to variations in detection and movement between the real-life recording and the simulation.

### 4.2. Comparison with Cross-Correlation

In this section, we utilized cross-correlation to compare the activation and order of sensors in both real and synthetic log sequences. The cross-correlation values were calculated for all sensors at each time point to evaluate the similarity between the two datasets.
(1)$$\left(f*g\right)\left[n\right]=\sum_{m=-\infty }^{\infty }\bar{f(m)}\;g\left(m+n\right)$$

The cross-correlation Formula (Equation 1) for discrete functions was used, and the cross-correlation values were calculated for all sensors and times in the sequences. To ensure a fair comparison with the longer real log sequences, we expanded the cross-correlation tables for synthetic sensors by duplicating the last line since the sensor values do not change.

To determine the similarity between the real and synthetic log sequences, we multiplied the value of each sensor in the real sequence by the corresponding synthetic sensor value. If the values matched (e.g., both *ON* or both *OFF*), a score of 1 was assigned; otherwise, a score of −1 was assigned. This calculation was performed for each sensor at each time point, resulting in the final cross-correlation table. The score was computed as the sum of all cross-correlation values for the sensors at each time.

The percentage likelihood between the two log sequences was calculated using the following formula:$$\mathrm{Percentage}=\frac{\mathrm{Maximum}\;\mathrm{Score}}{(\mathrm{Number}\;\mathrm{of}\;\mathrm{Sensors}\;\mathrm{in}\;\mathrm{Reality}\times \mathrm{Number}\;\mathrm{of}\;\mathrm{Events}\;\mathrm{in}\;\mathrm{Reality})}\times 100$$

Table 9 presents the results obtained for the 15 processed scenarios, including the similarity percentage and the average similarity across scenarios.

The obtained percentages were generally above 70%, except for one case that yielded 54.73%. Upon closer examination, we identified that the SensFloor sensor could be activated and deactivated multiple times in the same room, leading to variations in the log sequences.

In conclusion, the cross-correlation analysis revealed that the synthetic log sequences exhibited a high level of similarity to the real sequences, with similarity percentages exceeding 70% and reaching up to 86.53%. This indicates that the digital twin approach allowed us to generate synthetic data that closely resembled the real-world data. Although variations were observed in some scenarios due to the presence sensors, the overall comparison demonstrated a remarkable alignment between the two datasets. These findings suggest that the synthetic data generated through the digital twin approach can be effectively utilized for various applications, including activity recognition algorithms. In the following Section 4.3, we will investigate whether this level of similarity is sufficient to achieve comparable performance using an activity recognition algorithm.

### 4.3. Activity Recognition

To evaluate the generated dataset and our proposed method, we conducted four different tests using the activity recognition algorithm proposed by Liciotti et al. [5], as described in Section 4.3.1. The experiment methods, which involve data preprocessing and algorithm training, are detailed in Section 4.3.2.

#### 4.3.1. The Activity Recognition Algorithm

To validate the synthetic log sequences, we implemented an activity recognition algorithm based on the work by Liciotti et al. [5].

We selected this state-of-the-art algorithm due to its exceptional ability to demonstrate high performance, simplicity, and effectiveness with small datasets. Our rationale is that if a simple algorithm can achieve satisfactory results, more advanced and complex algorithms are likely to perform even better. By establishing a solid baseline using this straightforward approach, we can effectively gauge the potential for improvement with more sophisticated methods.

Moreover, the simplicity of the chosen algorithm offers several advantages. It is easier to understand, implement, and interpret, making it accessible to a broader audience. Additionally, it reduces computational overhead, enabling faster experimentation and iteration.

The algorithm architecture consists of an Embedding layer, a Bi-Directional LSTM layer [58], and a Softmax layer for classification, as shown in Figure 6.

The Embedding layer in the activity recognition algorithm transforms sensor activations, represented as integer values, into vectors. These vectors capture the underlying patterns and relationships in the sensor data, allowing the algorithm to learn meaningful representations of the input data.

On the other hand, the Bi-Directional LSTM processes the transformed sensor data in both forward and backward directions. This bidirectional processing enables the LSTM to capture dependencies and temporal patterns from both past and future time steps, making it well-suited for handling temporal time series or sequential data, such as the sensor data collected over time in a smart home environment.

We adopted identical hyperparameters to those outlined in the original paper. This encompassed an Embedding size of 64, coupled with a Bi-Directional LSTM housing 64 neurons. Notably, our model boasts 70,922 trainable parameters, occupying less than 10 MB of memory. This characteristic obviates the necessity for contemplating network pruning or resorting to transfer learning methodologies, given the relatively compact nature of the model.

#### 4.3.2. Preprocessing and Training Details

During our activity recognition tests, we encountered the limitation of not considering the time of day directly within the Virtual Home environment. Managing time in a virtual setting proved challenging, leading us to rely on post-processing techniques to estimate action durations. However, these estimates may not accurately represent real-world timing, introducing potential discrepancies in the data.

As a result, distinguishing between different meal times (morning, noon, or evening) during eating and cooking activities becomes challenging for the algorithm. Additionally, the data recording with volunteers for the three scenarios (morning, noon, or evening) was not performed at the correct time of the day, as the recordings were conducted during the daytime. Despite instructing the volunteers to project themselves at specific times (morning, mid-day, or evening), the presence of natural daylight during the experiments introduced biases in the execution of activities by the volunteers. This daylight-related bias can impact the realism and accuracy of the generated dataset, potentially affecting the performance evaluation of the activity recognition algorithm.

To address this challenge, we opted to avoid the timestamps and group the specific “Eat…” labels together under a general label, “Eat”. This grouping included activities such as (1) “Eat breakfast”, (2) “Eat lunch”, and (3) “Eat dinner”. Similarly, we combined the various cooking activities into a single label, “Cook”, encompassing (1) “Cook breakfast”, (2) “Cook lunch”, and (3) “Cook dinner”.

For our experiments, we employed a refined subset of our ground truth dataset. This subset exclusively retained the replicated scenarios and subjects as in the Table 7, ensuring a focused and pertinent approach to our analysis.

For each experiment, we consistently split our training data into 80% for training and 20% for validation. Additionally, to prevent overfitting, we implemented an early stop method, which automatically stops the training process when the model’s performance on the validation set no longer improves.

Moreover, we used a best model checkpoint technique to save the model at the end of each training epoch. The best model is selected based on the highest sparse categorical accuracy achieved during training. This ensures that we retain the most optimal model, which exhibited the best accuracy on the validation set.

#### 4.3.3. Experiments 1 and 2: Leave-One-Subject-Out Cross Validations

The objective of this step is to assess the performance of the activity recognition algorithm on both real and synthetic data. To achieve this, we employed the leave-one-subject-out cross-validation method, which is a variation of leave-one-out cross-validation [59]. This method is commonly used for datasets with limited data or when the data are associated with distinct entities.

In this experimental phase, we carried out two distinct leave-one-subject-out cross validations: the first exclusively utilizing synthetic data and the second exclusively employing real data. Our dataset encompassed three subjects, thus the leave-one-subject-out cross validations encompassed designating a single subject for testing purposes, while the remaining two subjects were dedicated to training and validating the model. For instance, during one iteration, subject 9 and subject 7 data were employed for training and validating, while the data from subject 3 was reserved for testing.

During the training phase, we used 80% of the data from two subjects to train the algorithm, while the remaining 20% of their data served as validation for the early stop algorithm. We repeated this process by rotating the subject used for testing, allowing us to test the algorithm on each subject’s data independently. Table 10 details the partitioning of activity sequence samples into training, validation, and test sets.

Finally, we compared the results obtained from the synthetic and real leave-one-subject-out cross-validations to evaluate the algorithm’s performance when trained and tested on synthetic data versus real data, respectively.

The results of the two leave-one-subject-out cross validations are presented in Table 11 and Table 12. These results demonstrate that the algorithm can be trained with both real and synthetic data, yielding comparable outcomes. Notably, there is a high degree of similarity in the results for Subject “9”. However, for the other two subjects, we observe more differences between the results of synthetic and real data. Specifically, the performance, as measured by F1-score and balanced accuracy, is better for Subject “7” and “3” with synthetic data.

Examining the confusion matrices in Figure 7, we observe more activity confusions when using real data. This finding suggests that real data pose greater complexity for the algorithm, likely due to sensor recording errors in the real apartment.

In general, the confusion matrices reveal certain patterns. For example, activities such as “Enter Home” and “Leave Home” are often confused, which is logical since they trigger similar types of sensors (entrance floor, principal door, etc.). Similarly, “Bathe” and “Go To Toilet” activities show confusion, likely because one may wash their hands after using the toilet, and in our apartment, the toilet is located in the same room as the bathroom. “Reading” and “Watching TV” activities can also be easily confused as they occur in the same room and trigger similar types of sensors. Additionally, “Wash Dishes” and “Cook” activities, being in the same room, are occasionally confused by the algorithm.

Comparing the confusion matrices for real and synthetic data, we observe that the aforementioned confusions occur in both cases. For instance, the activity “Read” is not correctly recognized in both synthetic and real data for Subject “3”, although the scores are slightly higher for real data. This difference can be attributed to the richness of real data, which exhibits more sensor activations in the sequence. Moreover, the avatar’s trajectory in simulation can introduce sensor activations that are not correlated with the current activity. In contrast, during the real recordings, subjects pay attention to room transitions, while the avatar does not, resulting in sensor activations from other rooms that disrupt activity sequences.

In conclusion, despite the logical confusions made by the algorithm, the recognition results obtained are quite similar for real and synthetic data. The next subsection (Section 4.3.4) investigates the extent to which synthetic data can be effectively used for activity recognition in our digital twin.

#### 4.3.4. Experiment 3: One-to-One—Training on Synthetic Data and Testing on Real Data

In this experiment, we trained the model using synthetic data from a subject and tested the trained algorithm with the real data of the same subject (see details in Figure 8). The objective was to determine whether an algorithm trained on synthetic data could effectively recognize activities in real data.

In more detail, three models were trained, one for each subject using his own synthetic and his own real data. To train these models, 80% of the synthetic data was used, while the remaining 20% was used for validation during the training process (details in Table 13). A stratified partitioning method was used to create these the training and validation subsets. After training, each algorithm was tested using real data from the corresponding subject.

Analyzing Table 14 and Figure 9, we can initially observe that the synthetic data generated for each subject enabled training and recognition of activity sequences for the corresponding real datasets (one subject at a time). Subjects “9” and “7” achieved good performance in terms of Accuracy, Balanced Accuracy, and F1-score. Notably, subject “7” exhibited the best performance among the subjects. For these two subjects, the synthetic data appeared realistic enough to achieve activity recognition with an accuracy of over 70% for both subjects.

In contrast, subject “3” displayed the lowest performance. It seems that the synthetic data generated for this subject were insufficient to enable accurate activity recognition. The poor performance for this subject suggests that there are differences between the synthetic and real data. A closer examination of the real data for subject “3” reveals sequences that are interfered with by sensors triggering unrelated to the activity. For example, the presence sensor on the floor is regularly triggered in the living room while subject “3” is in the kitchen. This disturbance occurs due to a sensor malfunction, detecting a second presence in the living room. Such malfunctions are not anticipated or simulated in the synthetic data.

Additionally, we observed that the activity “Bathe” was not recognized for subject “9”, whereas it was recognized with 100% accuracy for subject “7”. Subject “3” had four activity classes out of ten that were not recognized. These results indicate that synthetic data can be used to train an algorithm and recognize activities in real data. However, relying solely on activity data from a single subject may not always be sufficient. Furthermore, the performance can be degraded by sensor malfunctions in real conditions, which can disrupt the activity recognition algorithm. Therefore, incorporating more data and variability into the training dataset is necessary to address these challenges.

#### 4.3.5. Experiment 4: Many-to-One—Training on Synthetic Data and Testing on Real Data

The objective of this experiment was to train one model using all synthetic data and then test it on the real data of each subject independently (see Figure 10). The purpose was to evaluate whether increasing the amount of data provided by the synthetic data from different subjects would improve the algorithm’s capabilities in recognizing daily activities. In the previous section, we observed that training an algorithm on synthetic data and testing it in real conditions can be successful. However, the quantity of training data was not very representative and still relatively close to the real data. The performance could be enhanced by presenting the algorithm with examples of different lifestyles within the same living environment, allowing for better generalization.

To train the algorithm, we merged all the synthetic data into one dataset. Then, 80% of this dataset was used to train the algorithms, and 20% for training validation. Finally, the algorithm was tested over each subject’s datasets. The results are shown in Table 15.

Table 15 demonstrates that the algorithm achieved higher classification scores (close to 80%) for all subjects compared to the previous experiment. Subject “7” maintained very similar performance to the previous experiment. However, subjects “9” and “3” showed notable improvement, particularly subject “3”, which had previously exhibited the worst results. Subject “3” experienced an increase in accuracy and balanced accuracy from 57.14% and 47.50% to 78.57% and 81.67%, respectively.

Furthermore, Table 15 and Figure 11 reveal that more activity classes were correctly identified. The introduction of additional synthetic data from other subjects within the same apartment led to improved classification performance. The contribution of data from different subjects introduced variability in the execution of activities, enabling the algorithm to better generalize and capture sensor behavior during activities. Having a diverse range of examples is crucial for training a deep learning algorithm.

In conclusion, by utilizing more synthetic data, the algorithm demonstrated increased performance in real conditions. The inclusion of behavioral variability from different subjects facilitated better generalization. This generalization resulted in significant improvements, particularly for subject “3”.

### 4.4. Summary

The experiments conducted in this section yielded valuable insights. The results demonstrated that the simulator has the ability to generate synthetic data that closely resemble real-world data. The activity recognition algorithm performed similarly on both synthetic and real data, indicating that training the algorithm solely on synthetic data can effectively recognize activities in real-world scenarios. Moreover, when the entire set of generated synthetic data was utilized, the algorithm’s performance improved for each subject. This improvement can be attributed to the increased variability and examples provided by the additional synthetic data, allowing the algorithm to better generalize and capture the behavior of sensors during different activities.

## 5. Conclusions

In this study, we have explored the potential of leveraging a digital twin concept to generate synthetic data for Human Activity Recognition (HAR) algorithms in the context of smart homes. Our primary objective was to bridge the gap between training data and real-world usage data, effectively addressing the challenges posed by the variability and sparsity of activity traces in smart home environments.

To achieve this, we introduced the VirtualSmartHome simulator, which enhanced the Virtual Home environment to support sensor-based data simulation. With this simulator, we successfully replicated a real smart apartment and generated synthetic data by modeling the behaviors of residents through avatars. The extensive evaluation and metric analysis revealed a significant similarity between the synthetic data generated by the simulator and the real data collected from volunteers in the smart apartment.

We then utilized the synthetic data to train a HAR algorithm, which demonstrated robust activity recognition performance on the real data, achieving an average F1 score of approximately 80%. Although this experiment was conducted on a relatively small dataset, the promising results underscore the viability of our approach.

However, we acknowledge the need for further in-depth discussion and analysis to gain deeper insights from the results. In future work, we intend to explore the limitations of our study, specifically focusing on the impact of data collection in a lab environment versus a real-world home and the significance of dataset size. Understanding these aspects is critical for assessing the generalizability and practical applicability of our proposed approach.

To achieve this, we plan to expand the experiment by generating more synthetic data from additional volunteers’ activities. Additionally, we aim to extend the evaluation to include a larger number of real smart houses, allowing for a more comprehensive assessment of our approach’s performance across diverse environments.

Furthermore, we will explore the possibility of integrating scenarios with multiple agents to enrich datasets with more complex situations. Additionally, we intend to augment the simulator with new sensing modalities such as audio sensors and radar systems and more modalities. These additions will not only enhance the realism of the synthetic data but also broaden the scope of activity recognition research within smart homes.

In our future work, we aim to investigate training the algorithm from scratch in a house without replicating the labeled activities of the final resident. Instead, we will solely utilize the activity scripts provided by our volunteers, enabling a more realistic and autonomous training process. By addressing these aspects in our future work, we aim to further validate and enhance the effectiveness of our approach for generating synthetic data and training HAR algorithms in smart homes.

By harnessing the concept of digital twins and generating realistic synthetic data, we effectively mitigate the challenges posed by limited datasets and the gap between training and real environment data, thereby enhancing the applicability of HAR algorithms in real-world smart home environments. Our study contributes to the development of a comprehensive tool and methodology for implementing digital twins in the context of smart homes, enabling the development of more effective ADL classification algorithms and ultimately improving the performance of smart home systems.

In conclusion, the results obtained from this work highlight the potential of digital twins in generating synthetic data and training HAR algorithms for smart homes. Our future research will focus on addressing the identified limitations and further validating the approach with larger and more diverse datasets from real smart homes. We firmly believe that our findings significantly contribute to the advancement of HAR technology, paving the way for more efficient and adaptive smart home systems.

## Figures and Tables

**Figure 1 sensors-23-07586-f001:**
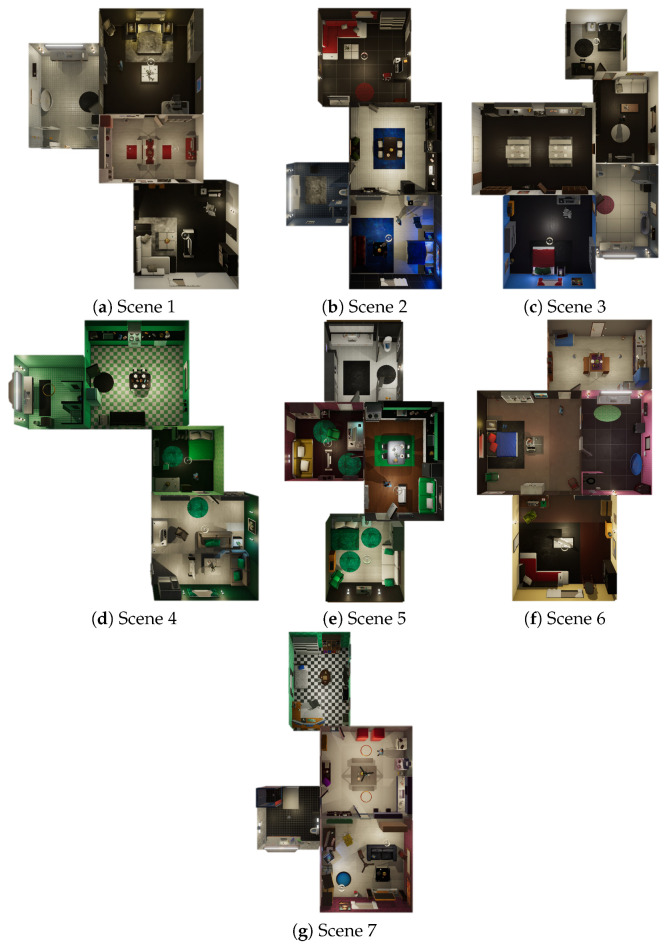
Virtual Home apartment scenes.

**Figure 2 sensors-23-07586-f002:**
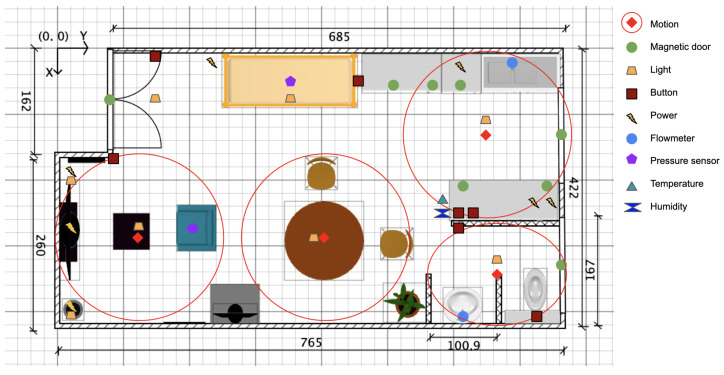
Layout showing the positions of the different sensors in the smart apartment.

**Figure 3 sensors-23-07586-f003:**
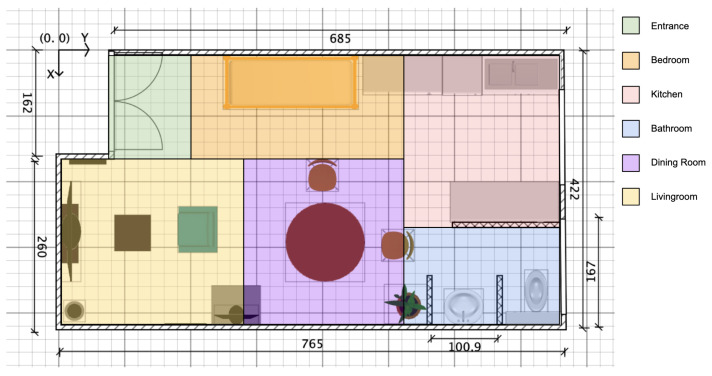
Room localization within the smart apartment.

**Figure 4 sensors-23-07586-f004:**
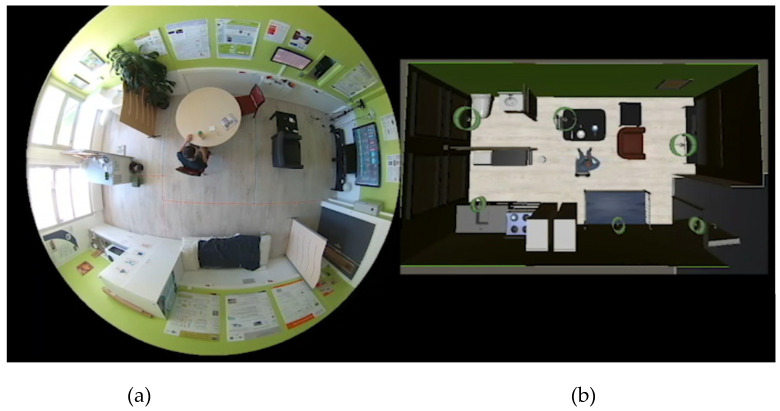
(**a**) View of the living lab from a fisheye camera, (**b**) representation in the Virtual Home simulator.

**Figure 5 sensors-23-07586-f005:**
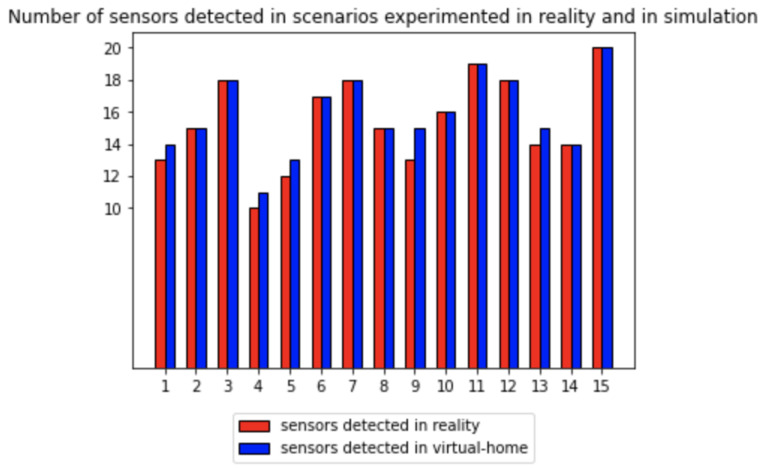
Comparison graph of sensors triggered in real and synthetic logs for similar scenarios.

**Figure 6 sensors-23-07586-f006:**
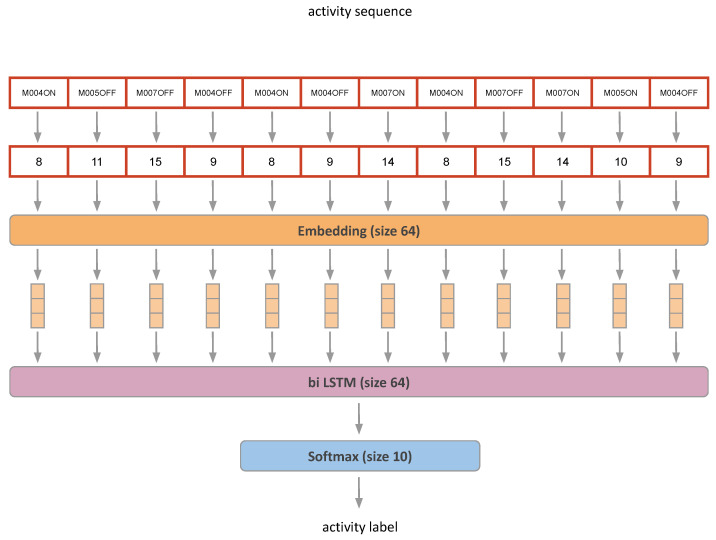
Bidirectional LSTM model architecture proposed by Liciotti et al. [5].

**Figure 7 sensors-23-07586-f007:**
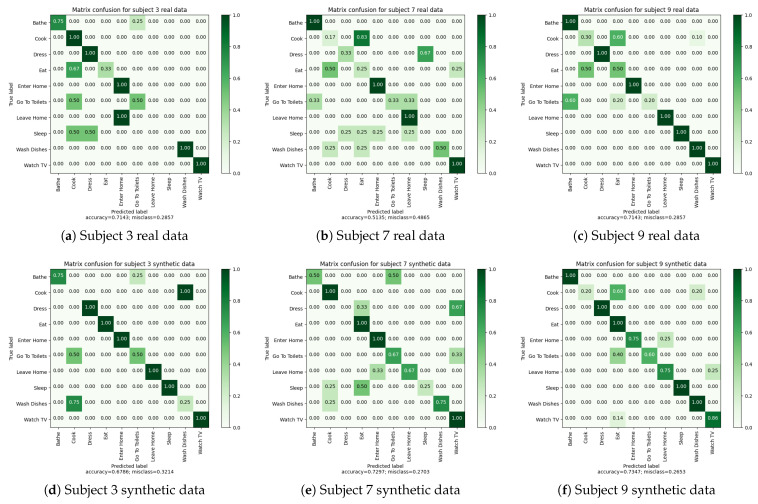
Experiments 1 and 2: Leave-one-subject-out cross-validations confusion matrices.

**Figure 8 sensors-23-07586-f008:**
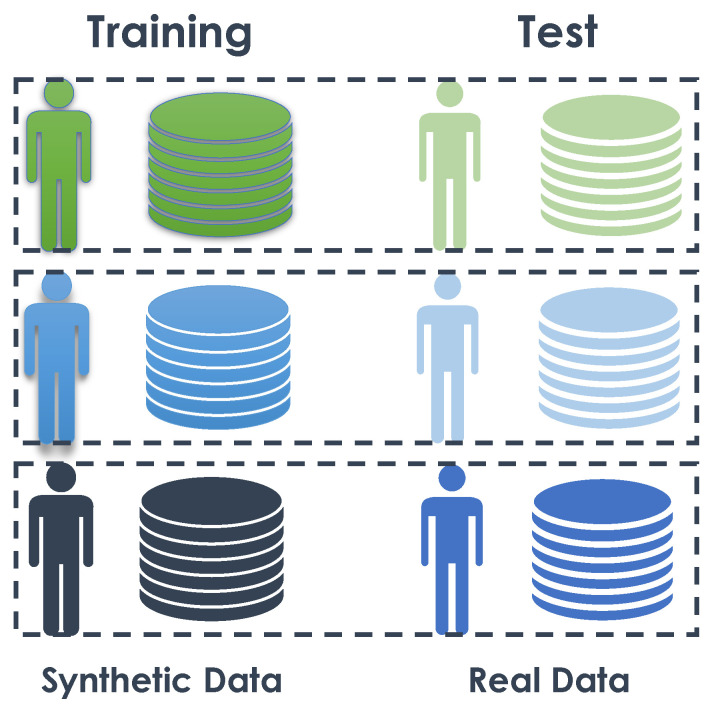
Experiment 3: One to one.

**Figure 9 sensors-23-07586-f009:**
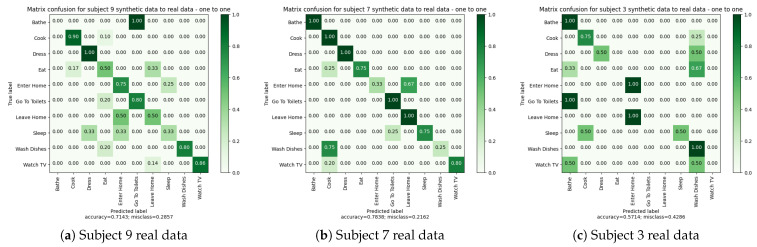
Experiment 3: One to one—confusion matrices.

**Figure 10 sensors-23-07586-f010:**
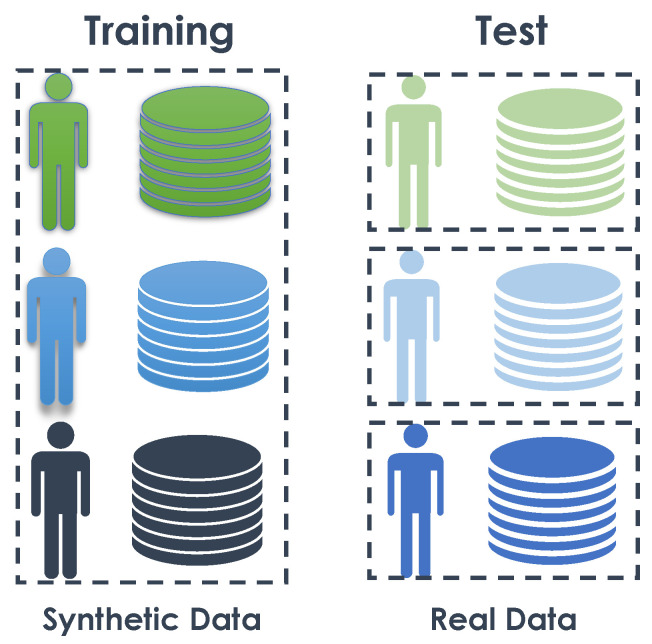
Experiment 4: Many to one.

**Figure 11 sensors-23-07586-f011:**
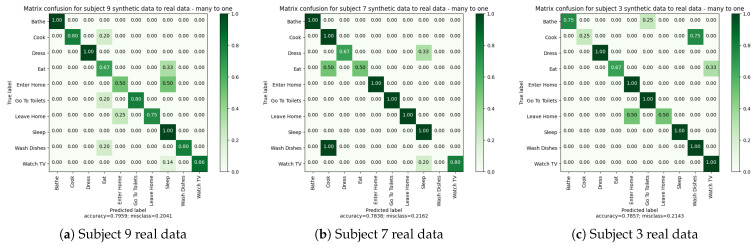
Experiment 4: Many to one—confusion matrices.

**Table 1 sensors-23-07586-t001:** Cost of CASAS components: “smart home in a box” [21].

Components	Server	Light and Motion Sensors	Door Sensors	Relay	Temperature Sensors	Total Cost
Unit Price	USD 350	USD 85	USD 75	USD 75	USD 75	
Quantity	1	24	1	2	2	
Total Price	USD 350	USD 2040	USD 75	USD 150	USD 150	**USD 2765**

**Table 2 sensors-23-07586-t002:** Comparison of simulation environments—table indicating for each simulator—its approach (model-based, interactive, or hybrid), its development environment, the language of the API, the number or name of the apartment recorded, its apartment designer/editor, its application, and its output (videos or sensor logs); whether it is open source, is multi-agent, records objects, uses activity scripts, provides a 3D visualisation; and whether it records IoT Sensors and their quantity.

Simulators	Open	Approach	Multi	Environment	API	Apartment	Objects	Scripts	IoT Sensors	Designer/Editor	Visual	Application	Output
AI2Thor [25]	Yes	Model	Yes	Unity	Python	17	609	No	No	Yes	3D	Robot Interaction	Videos
iGibson [26]	Yes	Model	Yes	Bullet Python	No	15	570	No	1	Yes	None	Robot Interaction	Videos
Sims4Action [27]	No	Model	Yes	Sims 4	No	None	NA	No	No	Game Interface	3D	Human Activity	Videos
Ai Habitat [28]	Yes	Model	Yes	C++	Python	None	NA	No	No	Yes	None	Human Activity	Sens. Log
Open SHS [29]	Yes	Hybrid	No	Blender	Python	None	NA	No	29 (B)	With Blender	3D	Human Activity	Sens. Log
SESim [30]	Yes	Model	No	Unity	NA	NA	Yes	Yes	5	Yes	3D	Human Activity	Sens. Log
Persim 3D [31]	No	Model	No	Unity	C#	Gator Tech	Yes	No	Yes (B)	Yes	3D	Human Activity	Sens. Log
IE Sim [32]	No	Hybrid	No	NA	NA	NA	Yes	No	Yes (B)	Yes	2D	Human Activity	Sens. Log
SIMACT [33]	Yes	Model	No	JME	Java	3D kitchen	Yes	Yes	Yes (B)	With Sketchup	3D	Human Activity	Sens. Log
Park et al. [34]	No	Interactive	No	Unity	NA	1	Yes	No	NA	With Unity	3D	Human Activity	Sens. Log
Francillette et al. [35]	Yes	Hybrid	Yes	Unity	NA	NA	Yes	Yes	8 (B and A)	With Unity	3D	Human Activity	Sens. Log
Buchmayr et al. [36]	No	Interactive	No	NA	NA	NA	Yes	No	Yes (B)	NA	3D	Human Activity	Sens. Log
Armac et al. [37]	No	Interactive	Yes	NA	NA	None	Yes	No	Yes (B)	Yes	2D	Human Activity	Sens. Log
VirtualHome [38]	Yes	Hybrid	Yes	Unity	Python	7	308	Yes	No	Yes	3D	Human Activity	Videos

**Table 3 sensors-23-07586-t003:** Summary of all recorded data from morning scenarios.

Activity	Subject 1	Subject 2	Subject 3	Subject 5	Subject 6	Subject 7	Subject 8	Subject 9	Total/Activity
**Bathe**	5	3	3	3	1	1	0	0	**16**
**Cook**	5	4	4	7	2	2	0	2	**26**
**Dress**	6	4	4	1	1	1	0	2	**19**
**Eat**	5	4	3	3	1	2	0	1	**19**
**Enter Home**	0	0	0	1	0	0	0	0	**1**
**Go To Toilets**	5	4	3	1	1	1	0	1	**16**
**Leave Home**	5	4	2	2	1	2	0	1	**17**
**Read**	0	0	2	2	0	0	0	0	**4**
**Sleep**	5	4	4	0	1	2	0	1	**17**
**Sleep in Bed**	0	0	0	2	0	0	0	0	**2**
**Wash Dishes**	5	4	3	0	1	2	0	1	**16**
**Watch TV**	0	0	0	3	1	0	0	2	**6**
**Total/Subject**	**41**	**31**	**28**	**25**	**10**	**13**	**0**	**11**	**159**

**Table 4 sensors-23-07586-t004:** Summary of all recorded data from mid-day scenarios.

Activity	Subject 1	Subject 2	Subject 3	Subject 5	Subject 6	Subject 7	Subject 8	Subject 9	Total/Activity
**Bathe**	5	0	1	3	0	0	0	0	**9**
**Cook**	7	6	1	3	1	4	2	5	**29**
**Dress**	7	2	0	1	0	0	0	1	**11**
**Eat**	5	2	1	3	0	2	1	2	**16**
**Enter Home**	5	2	1	2	1	2	1	2	**16**
**Go To Toilets**	8	4	1	4	1	0	0	2	**20**
**Leave Home**	5	1	1	2	1	2	1	2	**15**
**Read**	1	3	1	2	0	0	0	0	**7**
**Sleep**	2	0	0	1	0	0	0	0	**3**
**Sleep in Bed**	4	0	0	0	0	0	0	0	**4**
**Wash Dishes**	6	2	1	2	1	2	1	2	**17**
**Watch TV**	0	2	0	0	1	4	1	0	**8**
**Total/Subject**	**55**	**24**	**8**	**23**	**6**	**16**	**7**	**16**	**155**

**Table 5 sensors-23-07586-t005:** Summary of all recorded data from the evening scenarios.

Activity	Subject 1	Subject 2	Subject 3	Subject 5	Subject 6	Subject 7	Subject 8	Subject 9	Total/Activity
**Bathe**	6	2	3	5	0	1	0	2	**19**
**Cook**	6	5	2	5	0	3	0	3	**24**
**Dress**	8	2	1	4	1	2	1	0	**19**
**Eat**	4	2	2	4	0	1	0	2	**15**
**Enter Home**	5	2	2	3	1	2	1	2	**18**
**Go To Toilets**	10	3	1	4	0	2	0	2	**22**
**Leave Home**	0	0	0	0	0	0	0	0	**0**
**Read**	5	1	0	0	0	1	0	0	**7**
**Sleep**	4	2	1	3	1	2	1	2	**16**
**Sleep in Bed**	5	2	3	3	1	2	1	2	**19**
**Wash Dishes**	6	2	2	2	0	1	0	2	**15**
**Watch TV**	4	3	2	7	1	3	1	5	**26**
**Total/Subject**	**63**	**26**	**19**	**40**	**5**	**20**	**5**	**22**	**200**

**Table 6 sensors-23-07586-t006:** Summary of all recorded data from all scenarios.

Activity	Subject 1	Subject 2	Subject 3	Subject 5	Subject 6	Subject 7	Subject 8	Subject 9	Total/Activity
**Bathe**	16	5	7	11	1	2	0	2	**44**
**Cook**	18	15	7	15	3	9	2	10	**79**
**Dress**	21	8	5	6	2	3	1	3	**49**
**Eat**	14	8	6	10	1	5	1	6	**50**
**Enter Home**	10	4	3	6	2	4	2	4	**35**
**Go To Toilets**	23	11	5	9	2	3	0	5	**58**
**Leave Home**	10	5	3	4	2	4	1	4	**32**
**Read**	6	4	3	4	0	1	0	0	**18**
**Sleep**	11	6	5	4	2	4	1	2	**36**
**Sleep in Bed**	9	2	3	5	1	2	1	1	**25**
**Wash Dishes**	17	8	6	4	2	5	1	5	**48**
**Watch TV**	4	5	2	10	3	7	2	7	**40**
**Total/Subject**	**159**	**81**	**55**	**88**	**21**	**49**	**12**	**49**	**514**

**Table 7 sensors-23-07586-t007:** Synthétique Dataset Details.

Activity	Bathe	Cook	Dress	Eat	Enter Home	Go To Toilet	Leave Home	Sleep	Wash Dishes	Watch TV	Total/Subject
**Subject 3**	4	4	2	3	3	2	2	2	4	2	**28**
**Subject 7**	2	6	3	4	3	3	3	4	4	5	**37**
**Subject 9**	2	10	3	6	4	5	4	3	5	7	**49**
**Total/Activity**	**8**	**20**	**8**	**13**	**10**	**10**	**9**	**9**	**13**	**14**	**114**

**Table 8 sensors-23-07586-t008:** Comparison table of sensors triggered in real and synthetic logs.

Scenario Number	Real	Synthetic	Excess Sensors Detected	Index in Figure 5
s3_1_bis	13	14	floor_bedroom	1
s3_2_bis	15	15		2
s3_3_bis	18	18		3
s3_4_bis	10	11	floor_dining_room	4
s7_1	12	13	floor_bathroom	5
s7_2	17	17		6
s7_4	18	18		7
s7_5	15	15		8
s7_6	13	15	floor_bathroom, floor_bedroom	9
s9_1	16	16		10
s9_2	19	19		11
s9_3	18	18		12
s9_4	14	15	floor_bathroom	13
s9_5	14	14		14
s9_6	20	20		15

**Table 9 sensors-23-07586-t009:** Cross correlation similarity.

Subject	S9						S7					S3			
Scenario Index	1	2	3	4	5	6	7	8	9	10	11	12	13	14	15
Similarity (%)	75.73%	97.40%	75.42%	75.30%	69.61%	74.42%	81.24%	93.18%	85.22%	84.73%	88.28%	79.93%	77.53%	54.73%	82.78%
Average Similarity (%)	77.98%						86.53%					73.74%			

**Table 10 sensors-23-07586-t010:** Experiments 1 and 2: Partitioning of activity sequence samples into training, validation, and test sets.

	# Train Samples	# Validation Samples	# Test Samples
Subject 3	68	18	28
Subject 7	61	16	37
Subject 9	52	13	49

**Table 11 sensors-23-07586-t011:** Experiment 1: Leave-one-subject-out cross-validations for real data.

	Subject 9				Subject 7				Subject 3			
	**Precision**	**Recall**	**F1-Score**	**Support**	**Precision**	**Recall**	**F1-Score**	**Support**	**Precision**	**Recall**	**F1-Score**	**Support**
Bathe	40.00%	100.00%	57.14%	2	66.67%	100.00%	80.00%	2	100.00%	75.00%	85.71%	4
Cook	50.00%	30.00%	37.50%	10	25.00%	16.67%	20.00%	6	50.00%	100.00%	66.67%	4
Dress	100.00%	100.00%	100.00%	3	50.00%	33.33%	40.00%	3	66.67%	100.00%	80.00%	2
Eat	30.00%	50.00%	37.50%	6	12.50%	25.00%	16.67%	4	100.00%	33.33%	50.00%	3
Enter Home	100.00%	100.00%	100.00%	4	75.00%	100.00%	85.71%	3	60.00%	100.00%	75.00%	3
Go To Toilets	100.00%	20.00%	33.33%	5	100.00%	33.33%	50.00%	3	50.00%	50.00%	50.00%	2
Leave Home	100.00%	100.00%	100.00%	4	60.00%	100.00%	75.00%	3	0.00%	0.00%	0.00%	2
Sleep	100.00%	100.00%	100.00%	3	0.00%	0.00%	0.00%	4	0.00%	0.00%	0.00%	2
Wash Dishes	83.33%	100.00%	90.91%	5	100.00%	50.00%	66.67%	4	100.00%	100.00%	100.00%	4
Watch TV	100.00%	100.00%	100.00%	7	83.33%	100.00%	90.91%	5	100.00%	100.00%	100.00%	2
Accuracy	71.43%				51.35%				71.43%			
Balanced Accuracy	80.00%				55.83%				65.83%			
Macro Avg	80.33%	80.00%	75.64%	49	57.25%	55.83%	52.50%	37	62.67%	65.83%	60.74%	28
Weighted Avg	77.07%	71.43%	70.11%	49	54.19%	51.35%	49.19%	37	68.33%	71.43%	65.88%	28

**Table 12 sensors-23-07586-t012:** Experiment 2: leave-one-subject-out cross-validations for synthetic data.

	Subject 9				Subject 7				Subject 3			
	**Precision**	**Recall**	**F1-Score**	**Support**	**Precision**	**Recall**	**F1-Score**	**Support**	**Precision**	**Recall**	**F1-Score**	**Support**
Bathe	100.00%	100.00%	100.00%	2	100.00%	50.00%	66.67%	2	100.00%	75.00%	85.71%	4
Cook	100.00%	20.00%	33.33%	10	75.00%	100.00%	85.71%	6	0.00%	0.00%	0.00%	4
Dress	100.00%	100.00%	100.00%	3	0.00%	0.00%	0.00%	3	100.00%	100.00%	100.00%	2
Eat	40.00%	100.00%	57.14%	6	57.14%	100.00%	72.73%	4	100.00%	100.00%	100.00%	3
Enter Home	100.00%	75.00%	85.71%	4	75.00%	100.00%	85.71%	3	100.00%	100.00%	100.00%	3
Go To Toilets	100.00%	60.00%	75.00%	5	66.67%	66.67%	66.67%	3	50.00%	50.00%	50.00%	2
Leave Home	75.00%	75.00%	75.00%	4	100.00%	66.67%	80.00%	3	100.00%	100.00%	100.00%	2
Sleep	100.00%	100.00%	100.00%	3	100.00%	25.00%	40.00%	4	100.00%	100.00%	100.00%	2
Wash Dishes	71.43%	100.00%	83.33%	5	100.00%	75.00%	85.71%	4	20.00%	25.00%	22.22%	4
Watch TV	85.71%	85.71%	85.71%	7	62.50%	100.00%	76.92%	5	100.00%	100.00%	100.00%	2
Accuracy	73.47%				72.97%				67.86%			
Balanced Accuracy	81.57%				68.33%				75.00%			
Macro Avg	87.21%	81.57%	79.52%	49	73.63%	68.33%	66.01%	37	77.00%	75.00%	75.79%	28
Weighted Avg	85.66%	73.47%	71.65%	49	73.41%	72.97%	68.19%	37	70.71%	67.86%	68.99%	28

**Table 13 sensors-23-07586-t013:** Experience 3: Partitioning of activity sequence samples into training, validation, and test sets.

	# Train Samples from Synthetic Data	# Validation Samples from Synthetic Data	# Test Samples from Real Data
Subject 3	18	10	28
Subject 7	27	10	37
Subject 9	39	10	49

**Table 14 sensors-23-07586-t014:** Experiment 3: One to one—results of multiple metrics.

	S9				S7				S3			
	**Precision**	**Recall**	**F1-Score**	**Support**	**Precision**	**Recall**	**F1-Score**	**Support**	**Precision**	**Recall**	**F1-Score**	**Support**
Bathe	0.00%	0.00%	0.00%	2	100.00%	100.00%	100.00%	2	50.00%	100.00%	66.67%	4
Cook	90.00%	90.00%	90.00%	10	54.55%	100.00%	70.59%	6	75.00%	75.00%	75.00%	4
Dress	75.00%	100.00%	85.71%	3	100.00%	100.00%	100.00%	3	100.00%	50.00%	66.67%	2
Eat	50.00%	50.00%	50.00%	6	100.00%	75.00%	85.71%	4	0.00%	0.00%	0.00%	3
Enter Home	50.00%	75.00%	60.00%	4	100.00%	33.33%	50.00%	3	60.00%	100.00%	75.00%	3
Go To Toilets	66.67%	80.00%	72.73%	5	75.00%	100.00%	85.71%	3	0.00%	0.00%	0.00%	2
Leave Home	40.00%	50.00%	44.44%	4	60.00%	100.00%	75.00%	3	0.00%	0.00%	0.00%	2
Sleep	50.00%	33.33%	40.00%	3	100.00%	75.00%	85.71%	4	100.00%	50.00%	66.67%	2
Wash Dishes	100.00%	80.00%	88.89%	5	100.00%	25.00%	40.00%	4	44.44%	100.00%	61.54%	4
Watch TV	100.00%	85.71%	92.31%	7	100.00%	80.00%	88.89%	5	0.00%	0.00%	0.00%	2
Accuracy	71.43%				78.38%				57.14%			
Balanced Accuracy	64.40%				78.83%				47.50%			
Macro Avg	62.17%	64.40%	62.41%	49	88.95%	78.83%	78.16%	37	42.94%	47.50%	41.15%	28
Weighted Avg	70.78%	71.43%	70.39%	49	87.36%	78.38%	76.91%	37	44.92%	57.14%	46.59%	28

**Table 15 sensors-23-07586-t015:** Experiment 3: Many to one—results of multiple metrics.

	S9				S7				S3			
	**Precision**	**Recall**	**F1-Score**	**Support**	**Precision**	**Recall**	**F1-Score**	**Support**	**Precision**	**Recall**	**F1-Score**	**Support**
Bathe	100.00%	100.00%	100.00%	2	100.00%	100.00%	100.00%	2	100.00%	75.00%	85.71%	4
Cook	100.00%	80.00%	88.89%	10	50.00%	100.00%	66.67%	6	100.00%	25.00%	40.00%	4
Dress	100.00%	100.00%	100.00%	3	100.00%	66.67%	80.00%	3	100.00%	100.00%	100.00%	2
Eat	50.00%	66.67%	57.14%	6	100.00%	50.00%	66.67%	4	100.00%	66.67%	80.00%	3
Enter Home	66.67%	50.00%	57.14%	4	100.00%	100.00%	100.00%	3	75.00%	100.00%	85.71%	3
Go To Toilets	100.00%	80.00%	88.89%	5	100.00%	100.00%	100.00%	3	66.67%	100.00%	80.00%	2
Leave Home	100.00%	75.00%	85.71%	4	100.00%	100.00%	100.00%	3	100.00%	50.00%	66.67%	2
Sleep	37.50%	100.00%	54.55%	3	66.67%	100.00%	80.00%	4	100.00%	100.00%	100.00%	2
Wash Dishes	100.00%	80.00%	88.89%	5	0.00%	0.00%	0.00%	4	57.14%	100.00%	72.73%	4
Watch TV	100.00%	85.71%	92.31%	7	100.00%	80.00%	88.89%	5	66.67%	100.00%	80.00%	2
Accuracy	79.59%				78.38%				78.57%			
Balanced Accuracy	81.74%				79.67%				81.67%			
Macro Avg	85.42%	81.74%	81.35%	49	81.67%	79.67%	78.22%	37	86.55%	81.67%	79.08%	28
Weighted Avg	87.33%	79.59%	81.67%	49	77.48%	78.38%	74.89%	37	86.44%	78.57%	76.58%	28

## Data Availability

Not applicable.

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
