# Peer review of "A Smart Home Digital Twin to Support the Recognition of Activities of Daily Living"

_sensors, 2023, doi:10.3390/s23177586_

Round 1

Reviewer 1 Report

1. A separate section for the proposed neural network model for object classification is required. Also, please provide a figure to show how the neural network is composed layer by layer.

2. Some advanced techniques to reduce the computational complexity and memory usage are recently available, such as weight quantization, network pruning, etc.

[1] Z Tang, et al., "Automatic Sparse Connectivity Learning for Neural Networks", IEEE Transactions on Neural Networks and Learning Systems, 2022.

[2] Q. Huang, et al., "Weight-quantized squeezenet for resource-constrained robot vacuums for indoor obstacle classification", AI, 2022.

[3] C. Gong, et al., "VecQ: Minimal Loss DNN Model Compression With Vectorized Weight Quantization", IEEE Transactions on Computers, Volume 70, Issue 5, 01 May 2021.

3. In this study, only a limited number of data samples are collected for experimental study. You may consider to use transfer learning technique to increase the generalization ability of your trained model. Such as

J. Zheng, et al., "Improving the Generalization Ability of Deep Neural Networks for Cross-Domain Visual Recognition", IEEE Transactions on Cognitive and Developmental Systems 13 (3), 607-620, 2021.

F. Zhuang, et al., "A comprehensive survey on transfer learning", Proceedings of the IEEE, Volume: 109, issue 1, January 2021.

acceptable

Author Response

1. A separate section for the proposed neural network model for object classification is required. Also, please provide a figure to show how the neural network is composed layer by layer.

We have addressed the concern by adding more details about the model design in Section 4.4.1 to provide a clearer and more rigorous explanation of its usage in the experimental setup.

2. Some advanced techniques to reduce the computational complexity and memory usage are recently available, such as weight quantization, network pruning, etc.

[1] Z Tang, et al., "Automatic Sparse Connectivity Learning for Neural Networks", IEEE Transactions on Neural Networks and Learning Systems, 2022.

[2] Q. Huang, et al., "Weight-quantized squeezenet for resource-constrained robot vacuums for indoor obstacle classification", AI, 2022.

[3] C. Gong, et al., "VecQ: Minimal Loss DNN Model Compression With Vectorized Weight Quantization", IEEE Transactions on Computers, Volume 70, Issue 5, 01 May 2021.

The neural network model we used in our study is relatively small, and the amount of data currently available in our field does not necessitate model compression or optimization. Unlike larger models in computer vision or natural language processing fields, our model's size and computational complexity are not a significant concern.

Furthermore, the focus of our study is on evaluating a method to prepare an activity recognition model for a new environment. Given the limited amount of data available for our specific application domain, our primary goal is to address the challenges of variability and sparsity in activity traces within smart home environments. As such, model compression or advanced optimization techniques, such as weight quantization or network pruning, were not considered in this study.

While the cited techniques may indeed be beneficial for resource-constrained applications or large-scale data scenarios, they are currently beyond the scope of our investigation. Our objective was to demonstrate the effectiveness of the proposed approach using a digital twin concept and synthetic data generation for activity recognition in smart homes. Future research may explore the application of such techniques as the field progresses and more data becomes available to further optimize HAR models.

3. In this study, only a limited number of data samples are collected for experimental study. You may consider to use transfer learning technique to increase the generalization ability of your trained model. Such as

J. Zheng, et al., "Improving the Generalization Ability of Deep Neural Networks for Cross-Domain Visual Recognition", IEEE Transactions on Cognitive and Developmental Systems 13 (3), 607-620, 2021.

F. Zhuang, et al., "A comprehensive survey on transfer learning", Proceedings of the IEEE, Volume: 109, issue 1, January 2021.

Thank you for the suggestion. However, our study is focused on activity recognition in smart homes, and our dataset and research scope differ from the cross-domain visual recognition tasks typically addressed in transfer learning studies for images or videos. Our primary goal is to explore the potential of using a digital twin concept to generate synthetic data for activity recognition algorithms in the context of smart homes, with a focus on bridging the gap between training data and real-world usage data. While transfer learning is a valuable technique in various domains, it is currently not within the scope of our investigation.

Nonetheless, we appreciate your input, and we will consider the relevance of transfer learning in our future research, especially as our dataset and research area continue to evolve. We welcome further discussions and collaborations in exploring advanced techniques for improving the generalization ability of our trained models in the context of activity recognition for smart homes.

Reviewer 2 Report

The inherent and evolving diversity of users and their needs, home layouts and technologies challenge the development of smart homes (SH), particularly in terms of generalization and personalization.

The presented manuscript concerns the problem of variability and sparsity in recorded data in SH and how it challenges the development of robust human activity recognition systems (HAR).

To address the problem, the authors propose using digital twins and a Virtual Smart Home simulator to generate more varied datasets and reduce the disparity between training and real-world data.

The literature review includes an overview of a few existing methods employed in HAR and available smart home datasets. The literature review also includes comprehensive coverage, including limitations, of existing smart home simulators used to generate synthetic data.

The proposed solution is the Virtual Smart Home simulator, adapted from the VirtualHome simulator. It models ADLs in SH using interactive objects, simulated IoT sensors, and a user interface. It also enables replicating real-life apartments, simulation time acceleration, and logs.

To demonstrate and evaluate the Virtual Smart Home simulator's capacity to generate real data for training HAR algorithms, the authors compared data generated by the simulator with the ground truth real-world data. However, the ground truth dataset is limited to 3 scenarios instead of recording 24h.

To test the effectiveness of synthetic data for HAR, the authors trained the model using synthetic data and tested it on actual data. The performance varied among subjects, and sensor malfunctions in real conditions and differences between synthetic and real data were attributed as factors affecting recognition accuracy.
According to the authors, synthetic data showed promise for training HAR algorithms, but incorporating more data and variability is necessary to address challenges posed by sensor malfunctions and individual differences in behaviour.

The manuscript is well-written, with a clear, logical flow of ideas, starting with the challenges concerning HAR in SH, limitations of existing simulators, and the solution.

The manuscript is well-written, with a clear, logical flow of ideas, starting with the challenges concerning HAR in SH, limitations of existing simulators, and the solution.
Remarks:
- Line 244: unexpected new line.
- Repeated sections:
  - 3.2. Assessing the Simulator through Dataset Creation
  - 3.3. Assessing the Simulator through Dataset Creation

Author Response

The inherent and evolving diversity of users and their needs, home layouts and technologies challenge the development of smart homes (SH), particularly in terms of generalization and personalization.

The presented manuscript concerns the problem of variability and sparsity in recorded data in SH and how it challenges the development of robust human activity recognition systems (HAR).

To address the problem, the authors propose using digital twins and a Virtual Smart Home simulator to generate more varied datasets and reduce the disparity between training and real-world data.

The literature review includes an overview of a few existing methods employed in HAR and available smart home datasets. The literature review also includes comprehensive coverage, including limitations, of existing smart home simulators used to generate synthetic data.

The proposed solution is the Virtual Smart Home simulator, adapted from the VirtualHome simulator. It models ADLs in SH using interactive objects, simulated IoT sensors, and a user interface. It also enables replicating real-life apartments, simulation time acceleration, and logs.

To demonstrate and evaluate the Virtual Smart Home simulator's capacity to generate real data for training HAR algorithms, the authors compared data generated by the simulator with the ground truth real-world data. However, the ground truth dataset is limited to 3 scenarios instead of recording 24h.}

To test the effectiveness of synthetic data for HAR, the authors trained the model using synthetic data and tested it on actual data. The performance varied among subjects, and sensor malfunctions in real conditions and differences between synthetic and real data were attributed as factors affecting recognition accuracy.

According to the authors, synthetic data showed promise for training HAR algorithms, but incorporating more data and variability is necessary to address challenges posed by sensor malfunctions and individual differences in behaviour.

The manuscript is well-written, with a clear, logical flow of ideas, starting with the challenges concerning HAR in SH, limitations of existing simulators, and the solution.

- Line 244: unexpected new line.

We apologize for the formatting issue and will ensure that the new line is removed in the final version.

- Repeated sections:
  - 3.2. Assessing the Simulator through Dataset Creation
  - 3.3. Assessing the Simulator through Dataset Creation

We apologize for the oversight. The repeated sections have been removed in the revised version of the manuscript.

Reviewer 3 Report

In this article, the authors present a simulation environment for generating synthetic datasets to recognize daily activities in smart homes. The article is well-structured and addresses a relevant problem. However, there are a few issues that need to be resolved before publication.

  1. In the title, the authors use the term "Digital Twin”. Throughout the article, this term is often used as a synonym for a simulator. However, "Digital Twin" has a precise connotation and is used to describe the interplay of a system in the real world with its replica in a digital world. A Digital Twin includes simulation aspects, but they are used inside a bidirectional stream of information between the virtual and the real world. In this article, this characteristic is absent, and it seems that what the authors describe is only a simulator. Therefore, I strongly suggest that the authors reconsider the adoption of the "Digital Twin" keyword in the title and throughout the text.

  2. At the end of section 1.1, the phrase “Therefore, algorithms …. that encompasses this wide variability” suggests that the system proposed in the article is only useful for machine learning approaches. However, as the authors may know, the literature is rich with approaches for ADL recognition that do not rely on machine learning. Will their work be helpful for those approaches as well? I believe the answer is yes, and I suggest that the authors not ignore the potential contribution of their work. Instead, if the authors believe that their work is only useful for machine learning approaches, I suggest that they clearly specify this in the article title.

  3. In section 1.3, the paper presents its contributions. The simulation platform, which is the main contribution of the work (VirtualSmartHome), is introduced as an extension of a pre-existing platform (Virtual Home). I believe this should also be made clear in the abstract.

  4. In Section 2.2 “Activities of Daily Leaving Datasets”, I think the article could benefit from a more precise description of the type of sensors usually adopted for ADL recognition. This would help the authors to support why they chose to integrate into the simulation some sensors and not others.

  5. At lines 244/245, there is an extra new line.

  6. Throughout the text, there is inconsistency regarding the naming of Virtual Home. Virtual Home, VirtualHome, and Virtual-Home are all used interchangeably. I suggest that the authors choose one version and use it consistently.

  7. In Figure 1, the images of the 7 scenes are too small. The authors could increase the image size and remove the white space to make this figure more readable.

  8. In Section 3.3.1, when presenting the six smart sensors in the simulated apartment, it would be helpful to connect these sensors with the five sensors "types" previously mentioned in lines 379-380. Otherwise, it may be unclear why the IoT sensors integrated in the Virtual Home are limited to five, while there are six sensors in the simulation.

  9. The authors need to report more details about their subjects. Age range, sex, selection criteria? Furthermore, why the subject with ID 4 does not exist?

  10. To me is not clear how the synthetic dataset has been created to replicate the recorded scenarios. This point is crucial to determine if what follows in the article is coherent. The authors should provide a more clear description.

  11. It is not easy to determine, but it seems that the authors have confused the concepts of validation and testing. The article never describes the portion of the dataset used for testing the ML modules. I suggest that the authors revise the article considering the meaning of training, validating, and testing for data-driven approaches.

  12. Table 9 is confusing, I suggest the authors include column labels.

  13. In the literature review the authors mentioned that a criticality of current solutions is that they do not allow collecting data for multiple users in the same house at the same time. This aspect has not been tested by the described study. Does the proposed simulator allow to simulate multiple users or this limitation has not been addressed?

  14. The author focused on IoT sensors for ADL recognition. However, since other approaches also exist, I think they should discuss how other sensing solutions can be integrated into their simulator.

Overall, the article is well-written.

Author Response

In this article, the authors present a simulation environment for generating synthetic datasets to recognize daily activities in smart homes. The article is well-structured and addresses a relevant problem. However, there are a few issues that need to be resolved before publication.

In the title, the authors use the term "Digital Twin”. Throughout the article, this term is often used as a synonym for a simulator. However, "Digital Twin" has a precise connotation and is used to describe the interplay of a system in the real world with its replica in a digital world. A Digital Twin includes simulation aspects, but they are used inside a bidirectional stream of information between the virtual and the real world. In this article, this characteristic is absent, and it seems that what the authors describe is only a simulator. Therefore, I strongly suggest that the authors reconsider the adoption of the "Digital Twin" keyword in the title and throughout the text.

We fully agree that the concept of the digital twin implies a kind of two-way flow of knowledge between the physical and digital worlds.
In the introduction, we set out our vision of the digital twin in the context of human activity recognition.
A digital version of the real habitat is modeled, intensive simulations are performed on it to train the algorithms, and the algorithms are used in the real habitat. If there are any changes to the real habitat, they can be fed back into the digital version and start the process again. Knowledge flows from one world to another.
It is true that this document does not provide all the links in the complete chain of this digital twin approach.
For example, how a real habitat is modeled digitally is not covered; neither is the way a HAR algorithm is run in real time at home. It deals more specifically with simulation.
However, we strongly believe that the solution described in this article fits into and contributes to the concept of a digital twin for the HAR.

Here we have information that flows in both directions but on a different time scale: (1) from a physical home, we create a digital model, (2) generate synthetic data for learning, and finally (3) transfer the knowledge from the synthetic data to perform inference in the real home.

At the end of section 1.1, the phrase “Therefore, algorithms …. that encompasses this wide variability” suggests that the system proposed in the article is only useful for machine learning approaches. However, as the authors may know, the literature is rich with approaches for ADL recognition that do not rely on machine learning. Will their work be helpful for those approaches as well? I believe the answer is yes, and I suggest that the authors not ignore the potential contribution of their work. Instead, if the authors believe that their work is only useful for machine learning approaches, I suggest that they clearly specify this in the article title.

Thank you for raising this important point. Our work has indeed been developed and evaluated in the context of machine learning approaches for activity recognition in smart homes. While the literature does encompass various approaches for ADL recognition that do not rely on machine learning, our focus in this study was specifically on the utilization of digital twins and synthetic data generation to train HAR algorithms.

However, you are correct that the proposed approach can be potentially applicable to other ADL recognition methods that not rely on machine learning. We agree that acknowledging this potential contribution would be beneficial. 
We exposed this in the introduction.
We appreciate your feedback and value the opportunity to highlight the versatility of our proposed solution in accommodating different ADL recognition approaches.

In section 1.3, the paper presents its contributions. The simulation platform, which is the main contribution of the work (VirtualSmartHome), is introduced as an extension of a pre-existing platform (Virtual Home). I believe this should also be made clear in the abstract.

We have revised the abstract to explicitly mention that the main contribution of the work is the extension of the pre-existing platform, Virtual Home, to create the VirtualSmartHome simulator. However, it's important to note that our contributions are not solely limited to the simulator itself, but also include the methodology of applying the digital twin concept to generate training data that closely resembles the final environment.

We appreciate your feedback and have made the necessary changes to the abstract to accurately reflect the contributions of our work. Thank you for helping us improve the clarity of our paper.

In Section 2.2 “Activities of Daily Leaving Datasets”, I think the article could benefit from a more precise description of the type of sensors usually adopted for ADL recognition. This would help the authors to support why they chose to integrate into the simulation some sensors and not others.

In response to your comment, we have added a more precise description of the type of sensors typically adopted for ADL recognition in Section 2.2. This additional information will help support our decision-making process in integrating specific sensors into the simulation while excluding others. We hope this clarification will enhance the understanding of our approach and rationale for sensor selection.

At lines 244/245, there is an extra new line.

We apologize for the formatting issue and will ensure that the new line is removed in the final version.

Throughout the text, there is inconsistency regarding the naming of Virtual Home. Virtual Home, VirtualHome, and Virtual-Home are all used interchangeably. I suggest that the authors choose one version and use it consistently.

Thank you for bringing this inconsistency to our attention. We apologize for any confusion caused by the varying naming conventions. We have now unified the naming of "Virtual Home" throughout the text, using "Virtual Home" consistently. This will ensure clarity and consistency in the presentation of our work. We appreciate your valuable feedback and strive to improve the readability of our paper.

In Figure 1, the images of the 7 scenes are too small. The authors could increase the image size and remove the white space to make this figure more readable.

We have taken your feedback into consideration and made the necessary adjustments to Figure 1. The image size of the 7 scenes has been increased, and we have removed unnecessary white spaces between the subfigures to improve readability. We hope these changes enhance the clarity and presentation of our work. 

In Section 3.3.1, when presenting the six smart sensors in the simulated apartment, it would be helpful to connect these sensors with the five sensors "types" previously mentioned in lines 379-380. Otherwise, it may be unclear why the IoT sensors integrated in the Virtual Home are limited to five, while there are six sensors in the simulation.

Indeed the smart apartment is equipped with motion sensors that we didn't implement in Virtual Home. We also removed these sensors information in our real logs to be consistant. To clarify this, we have revised the text in Section 3.3.1.

The authors need to report more details about their subjects. Age range, sex, selection criteria? Furthermore, why the subject with ID 4 does not exist?

We have added some details in Section 3.2.2 about the age range and sex distribution of the participants. The age range of the participants was between 18 and 38 years. Regarding sex, we had an equal representation of 8 men and 3 women participants.

The selection criteria for the participants were based on their willingness to participate and their availability during the data collection period. We ensured that the participants were familiar with the smart home environment and had no mobility issues that could affect the accuracy of the activity recognition.

As for subject ID 4, unfortunately, they had to cancel their participation in the study due to personal reasons. Consequently, we only had data from subjects 1, 2, 3, 5, 6, and 7.

To me is not clear how the synthetic dataset has been created to replicate the recorded scenarios. This point is crucial to determine if what follows in the article is coherent. The authors should provide a more clear description.

We have revised the section 3.2.3 to provide a more detailed and clearer description of how the synthetic dataset was created to replicate the recorded scenarios. We hope that the added details will address your concerns and provide a better understanding of our approach.

It is not easy to determine, but it seems that the authors have confused the concepts of validation and testing. The article never describes the portion of the dataset used for testing the ML modules. I suggest that the authors revise the article considering the meaning of training, validating, and testing for data-driven approaches.

Thank you for bringing this to our attention. We apologize for any confusion caused by the confusion of concepts related to training, validation, and testing in our article. We have carefully revised the article and made the necessary adjustments to ensure clarity in the description of these processes.

To address this issue, we have added more details in each experiment section regarding the specific portions of the dataset used for training, validation, and testing of the machine learning modules. Additionally, we have included a new section (Section 4.4.2) that provides a comprehensive overview of the pre-processing and training methods used in our data-driven approaches.

Table 9 is confusing, I suggest the authors include column labels.

After careful consideration, we have decided to remove this table from the article. It contained an extract of our data, and we agree that its absence does not hinder the understanding of the article or the experiment.

In the literature review the authors mentioned that a criticality of current solutions is that they do not allow collecting data for multiple users in the same house at the same time. This aspect has not been tested by the described study. Does the proposed simulator allow to simulate multiple users or this limitation has not been addressed?

The simulator allows for the simulation of multiple agents. We have not conducted experiments in this specific configuration yet. However, if our results are validated in future works, we plan to conduct an experiment involving multiple agents.

The author focused on IoT sensors for ADL recognition. However, since other approaches also exist, I think they should discuss how other sensing solutions can be integrated into their simulator.

The literature indeed offers various methods for achieving human activity recognition, and there are other sensor types that could be integrated into our simulation. For instance, body skeleton data, such as from a Kinect camera, can be already extracted from the simulator, and this information could be used to simulate IMU sensors, among others. We have included a brief discussion about this topic in the conclusion section of the article.

Reviewer 4 Report

This paper identifies challenges in creating realistic datasets in smart home environments deploying ambient sensors to monitor the residents activities. As a result building effective supervised models is difficult. Key challenges include: the labelling of data; and variability in home layouts; sensor networks; and residents habits. This paper aims to mitigate these with simulated data generated from a digital twin.  

The paper is quite interesting. Simulated datasets aren't uncommon in smart home research, but expanding upon a popular platform for video-based smart home simulations to add IoT sensor support is novel and genuinely interesting.

There is extensive discussion on the approach, which is fine and clearly identifies the contributions of this paper compared to previously reported video-based simulations. The language is generally fine but a little casual in places describing step by step what has been done rather than a more academic overview of the work.

The related work is very good. Relevant work is cited and the section gives good additional context to this project.

There are extensive experimental results comparing the synthetic data to the real data. The 4 experiment on activity recognition with different designs and combinations of test / training data each reporting a table of results and individual confusion matrices is maybe slightly repetitive.Particularly when only one dataset and model are reported.

Performance appears only OK and there are no clear benchmarks by which to make comparisons. One complaint with the paper is that there is very little detail on the model design used in the experimental setup. The model is from a cited paper, however the reasons for using it are not clear. Given the fairly extensive review of model structures in the related work, it seems the reasoning could be a bit more well-defined here or an alternative model also included. Currently it is hard to speculate on how the model design impacted the activity recognition performance. This is important given the results are being used to demonstrate the viability of this approach for simulated dataset generation. 

A further issue with the paper is a lack of in-depth discussion. What was done and associated results are reported in detail. However, the reader is generally left to draw their own insights as to what they mean. There is little discussion on the limitations of the work, for example: does the data being captured from volunteers in a lab environment rather than a  real world home impact the results; or is the demographics of the participants relevant? I would also like to see some reflection on achievement and the extent to which the work here does mitigate the challenges identified at the start of the paper.  

Overall an interesting paper that people will find interesting. 

Minor Issues:

Typo - p19: "Figure 6. Experiment 1 an 2:"

Author Response

This paper identifies challenges in creating realistic datasets in smart home environments deploying ambient sensors to monitor the residents activities. As a result building effective supervised models is difficult. Key challenges include: the labelling of data; and variability in home layouts; sensor networks; and residents habits. This paper aims to mitigate these with simulated data generated from a digital twin.

The paper is quite interesting. Simulated datasets aren't uncommon in smart home research, but expanding upon a popular platform for video-based smart home simulations to add IoT sensor support is novel and genuinely interesting.

There is extensive discussion on the approach, which is fine and clearly identifies the contributions of this paper compared to previously reported video-based simulations. The language is generally fine but a little casual in places describing step by step what has been done rather than a more academic overview of the work.

The related work is very good. Relevant work is cited and the section gives good additional context to this project.

There are extensive experimental results comparing the synthetic data to the real data. The 4 experiment on activity recognition with different designs and combinations of test / training data each reporting a table of results and individual confusion matrices is maybe slightly repetitive. Particularly when only one dataset and model are reported.

We acknowledge that there might be a sense of repetition in the extensive experimental results, especially considering the four experiments on activity recognition with various designs and combinations of test/train data, each accompanied by tables of results and individual confusion matrices. However, we chose to maintain this format to ensure clarity, highlight the specific objectives, and present the results of each experiment more comprehensively. Additionally, we have included additional details to facilitate a better understanding of the findings.

Performance appears only OK and there are no clear benchmarks by which to make comparisons. One complaint with the paper is that there is very little detail on the model design used in the experimental setup. The model is from a cited paper, however the reasons for using it are not clear. Given the fairly extensive review of model structures in the related work, it seems the reasoning could be a bit more well-defined here or an alternative model also included. Currently it is hard to speculate on how the model design impacted the activity recognition performance. This is important given the results are being used to demonstrate the viability of this approach for simulated dataset generation. 

We have addressed the concern by adding more details about the model design in Section 4.4.1 to provide a clearer and more rigorous explanation of its usage in the experimental setup. This additional information aims to better define the reasoning behind using this particular model and to shed light on its potential impact on the activity recognition performance.

A further issue with the paper is a lack of in-depth discussion. What was done and associated results are reported in detail. However, the reader is generally left to draw their own insights as to what they mean. There is little discussion on the limitations of the work, for example: does the data being captured from volunteers in a lab environment rather than a  real world home impact the results; or is the demographics of the participants relevant? I would also like to see some reflection on achievement and the extent to which the work here does mitigate the challenges identified at the start of the paper.

We have revised the conclusion to provide a more detailed and clearer discussion.

Overall an interesting paper that people will find interesting.

Typo - p19: "Figure 6. Experiment 1 an 2:"

Thank you for bringing this to our attention

Round 2

Reviewer 1 Report

1. Figure 6 is vague and cannot be read clearly. Please use high-resolution figures instead.

2. Since your model is not very large, please report the number of trainable parameters in your model, and how much memory do you need to run your model. Thus, you may justify that there is no need to consider network pruning or transfer learning techniques. It is better to add a paragraph to discuss why you don't need to consider these techniques.

3. Please report the number of training, validation and testing samples in your dataset for experiments.

acceptable

Author Response

1. Figure 6 is vague and cannot be read clearly. Please use high-resolution figures instead.

We have increased the size and resolution of Figure 6 and included additional parameter details in both the text and the figure.

2. Since your model is not very large, please report the number of trainable parameters in your model, and how much memory do you need to run your model. Thus, you may justify that there is no need to consider network pruning or transfer learning techniques. It is better to add a paragraph to discuss why you don't need to consider these techniques.

We have provided specific details about the model, including the number of parameters and memory requirements, in Section 4.4.1. Additionally, we have addressed the absence of network pruning or transfer learning techniques by explaining our model's modest size and memory demands, which render these techniques unnecessary. This justification is now included in the manuscript.

3. Please report the number of training, validation and testing samples in your dataset for experiments.

For each experiment, we have included tables that present a breakdown of the dataset, showing the number of samples used for training, validation, and testing. This information is now readily available in the revised manuscript.

Reviewer 3 Report

Dear Authors,

I want to thank you for your work in improving the article. I have carefully revised the new version of the paper and your answers to my previous comments. Although some adjustments have been made, I feel some of your answers did not result in changes to the article. Here are my updated observations:

  1. The answer to my second comment concludes with

“ We exposed this in the introduction. We appreciate your feedback and value the opportunity to highlight the versatility of our proposed solution in accommodating different ADL recognition approaches.”

However, I have noticed that the introduction has not been updated. The previous comment was generated due to a lack of clarity regarding the original introduction. Therefore, I suggest that the authors revise the introduction to make it clear that while the ML approach is just one possibility, their solution could also be applied to more traditional applications.

  1. In reference to my earlier comment about section 3.3.1, which is now 3.2.1, I have noticed that no modifications have been made despite the authors' assertion that the text has been revised.

  2. Regarding my previous comment about the distinction between training, validation, and testing, the article is now more comprehensible. However, I still have some uncertainties. Firstly, to ensure we are in agreement, I would like to explain what training, validation, and test signify to me:

    1. Training - the set of data used to fit the parameters.
    2. Validation - a set to determine the model fit while tuning the model hyperparameters. Usually adopted to prove over-fitting and implement early stop conditions.
    3. Test - the set used to provide an unbiased evaluation of the model.

    Notice that the Validation can not be used to provide an unbiased evaluation since it indirectly affects the training process.

    Having said that, what I would expect is. Divide the synthetic data into 3 sets training, validation, and test (the splits can be different but a common one is 70/15/15). Use training and validation sets to tune the model parameters and the test set to present the results on the synthetic dataset. After this, you should test the model on the real data and show the difference between test on real data vs test on synthetic.

    Instead according to what is reported in the article the results on the synthetic data are shown on the validation set.

    I think the author improved the explanation but from a methodological perspective, this should be improved. Furthermore, image 8 was helpful to get a better grasp on the problem but it is never referenced in the article.

    1. Finally regarding my last two comments:
      a. I think that the authors could mention the possibility of multi-agent simulation in conclusion
      b. although the conclusions have been revised there is no reference to sensing modalities different from IoT.

Author Response

Dear Reviewer,

I would like to express my gratitude for your valuable feedback, which has greatly contributed to the improvement of our article. We have carefully incorporated your suggestions and made revisions accordingly. We are confident that this new version aligns better with your recommendations and addresses your concerns. Your input has been instrumental in enhancing the quality of our work, and we appreciate your time and effort in reviewing our manuscript.

Thank you once again for your insightful comments.

1. The answer to my second comment concludes with

“ We exposed this in the introduction. We appreciate your feedback and value the opportunity to highlight the versatility of our proposed solution in accommodating different ADL recognition approaches.”

However, I have noticed that the introduction has not been updated. The previous comment was generated due to a lack of clarity regarding the original introduction. Therefore, I suggest that the authors revise the introduction to make it clear that while the ML approach is just one possibility, their solution could also be applied to more traditional applications.

We have rectified this oversight and have ensured that the introduction now accurately highlights the adaptability of our approach to various ADL recognition methods.

2. In reference to my earlier comment about section 3.3.1, which is now 3.2.1, I have noticed that no modifications have been made despite the authors' assertion that the text has been revised.

I apologize for the confusion regarding the section 3.2.1. While the content of this section remains unchanged, pertinent details to address your concern have been incorporated in section 3.2.3. I hope this adjustment aligns with your expectations.

3. Regarding my previous comment about the distinction between training, validation, and testing, the article is now more comprehensible. However, I still have some uncertainties. Firstly, to ensure we are in agreement, I would like to explain what training, validation, and test signify to me:

  1. Training - the set of data used to fit the parameters.
  2. Validation - a set to determine the model fit while tuning the model hyperparameters. Usually adopted to prove over-fitting and implement early stop conditions.
  3. Test - the set used to provide an unbiased evaluation of the model.

Notice that the Validation can not be used to provide an unbiased evaluation since it indirectly affects the training process.

Having said that, what I would expect is. Divide the synthetic data into 3 sets training, validation, and test (the splits can be different but a common one is 70/15/15). Use training and validation sets to tune the model parameters and the test set to present the results on the synthetic dataset. After this, you should test the model on the real data and show the difference between test on real data vs test on synthetic.

Instead according to what is reported in the article the results on the synthetic data are shown on the validation set.

I think the author improved the explanation but from a methodological perspective, this should be improved. Furthermore, image 8 was helpful to get a better grasp on the problem but it is never referenced in the article.

We are totally agree with explanation of training, validation, and testing sets. And we followed exactly this methods. We have now provided tables for each experiment, detailing the distribution of samples across the training, validation, and testing sets. This will enable a clearer understanding of our methodology. 

4. Finally regarding my last two comments:
a. I think that the authors could mention the possibility of multi-agent simulation in conclusion
b. although the conclusions have been revised there is no reference to sensing modalities different from IoT.

Your input regarding the possibility of multi-agent simulation and the inclusion of sensing modalities other than IoT is highly valuable. We have integrated these considerations into the conclusions section, addressing both aspects as you suggested.

Once again, we extend my sincere gratitude for your meticulous evaluation and constructive feedback. Your assistance has undoubtedly contributed to the refinement of our work.